# BeGin: Extensive Benchmark Scenarios and An Easy-to-use Framework for Graph Continual Learning

## Abstract

Continual Learning (CL) is the process of learning ceaselessly a sequence of tasks. Most existing CL methods deal with independent data (e.g., images and text) for which many benchmark frameworks and results under standard experimental settings are available. Compared to them, however, CL methods for graph data (graph CL) are relatively underexplored because of (a) the lack of standard experimental settings, especially regarding how to deal with the dependency between instances, (b) the lack of benchmark datasets and scenarios, and (c) high complexity in implementation and evaluation due to the dependency. In this paper, regarding (a) we define four standard incremental settings (task-, class-, domain-, and time-incremental) for node-, link-, and graph-level problems, extending the previously explored scope. Regarding (b), we provide 31 benchmark scenarios based on 20 real-world graphs. Regarding (c), we develop BeGin, an easy and fool-proof framework for graph CL. BeGin is easily extended since it is modularized with reusable modules for data processing, algorithm design, and evaluation. Especially, the evaluation module is completely separated from user code to eliminate potential mistakes. Regarding benchmark results, we cover $3\times$ more combinations of incremental settings and levels of problems than the latest benchmark.

## 1 Introduction

Continual Learning (CL), which is also known as lifelong learning and incremental learning, is the process of learning continuously a sequence of tasks. CL aims to retain knowledge from previous tasks (i.e., knowledge consolidation) to overcome the degradation of performance (i.e., catastrophic forgetting) on previous tasks. Recently, CL has received considerable attention because of its similarity to the development of human intelligence.

Most of the existing works for CL deal with independent data, such as images and text. For example, Shin et al. (2017) aim to learn a sequence of image-classification tasks where the domains of images vary with tasks. Ermis et al. (2022) aim to learn a sequence of text-classification tasks where the possible classes of text grow over tasks. For CL with independent data, many datasets (Goodfellow et al., 2013; Hsu et al., 2018; Lomonaco & Maltoni, 2017) and benchmarks (He & Sick, 2021; Lin et al., 2021; Lomonaco et al., 2021; Pham et al., 2021; Shin et al., 2017) have been provided.

CL naturally provides significant benefits for graph data in real-world applications since the data grow in size and diversity across various domains accompanied by the emergence of new tasks (refer to Appendix A for a specific real-world scenario). The majority of existing graph representation learning methods, however, are designed for fixed tasks, assuming that the input graph remains unchanged (Kipf & Welling, 2017; Velickovic et al., 2017). Although a number of methods exist to handle graph evolution over time (Xu et al., 2020; Rossi et al., 2020), the ability to incorporate new tasks, classes, and/or domains remains uncommon (Wang et al., 2020a; Zhou & Cao, 2021).

Despite its necessity, graph CL (Febrinanto et al., 2022; Zhou & Cao, 2021) has been relatively underexplored compared to CL with independent data, mainly due to the complexity caused by the dependency between instances. For example, in node classification, the class of a node is correlated not only with its features but also with its connections to other nodes, which may belong to other tasks, and their features. Major reasons for the relative unpopularity of graph CL include the scarcity of standard experimental settings.[1] For example, how to deal with changes in various dimensions (e.g., domains and classes) and the dependency between instances needs to be standardized. Moreover, the relative scarcity of benchmarking datasets and scenarios also limits its popularity.

---

[1]While there exist some settings commonly considered in multiple studies, their scope is restricted in relation to tasks, incremental settings, and application domains, as discussed in Appendix J.

Table 1: **Comparison of graph CL benchmarks.** Ours includes more incremental settings, levels of problems, and graph CL methods and metrics. See Appendix J for a more extensive comparison.

| | | | BEGIN (Proposed) | | | (Zhang et al., 2022b) | | | (Carta et al., 2021) | | |
|---|---|---|---|---|---|---|---|---|---|---|---|
| | Problem Level | | Node | Link | Graph | Node | Link | Graph | Node | Link | Graph |
| Scenarios | Incremental Setting | Task | ✔ | ✔ | ✔ | ✔ | ✗ | ✔ | ✗ | ✗ | ✗ |
| | | Class | ✔ | ✔ | ✔ | ✔ | ✗ | ✔ | ✗ | ✗ | ✔ |
| | | Domain | ✔ | ✔ | ✔ | ✗ | ✗ | ✗ | ✗ | ✗ | ✗ |
| | | Time | ✔ | ✔ | ✔ | ✗ | ✗ | ✗ | ✗ | ✗ | ✗ |
| # of Graph CL Methods | | | **10** | | | 6 | | | 3 | | |
| # of Evaluation Metrics | | | **4** | | | 2 | | | 1 | | |

In this paper, we focus on resolving these issues. Our first contribution is to define four incremental settings for graph data, enriching those established in general CL but partially explored for Graph CL. To this end, we identify and decouple four possible dimensions of change, which are task, class, domain, and time. It is worth noting that previous studies often couple these dimensions of change complicatedly without sufficient justification (Zhang et al., 2022b; Zhou & Cao, 2021) (see Section 3.2 for details). Thus, there has been a lack of opportunities to consider each dimension on its own for effective algorithm design, evaluation, and other purposes. Each of the settings is defined so that the dependency between instances, which is a unique property of graph data, can be utilized. Simultaneously, each setting may lead to catastrophic forgetting without careful consideration of it. Then, we show that the settings can be applied to node-, link-, and graph-level problems, including node classification, link prediction, and graph classification. After that, we provide 31 benchmark scenarios for graphs from 20 real-world datasets, which cover 12 combinations of the incremental settings and the levels of problems, as summarized in Table 1 and discussed in detail in Appendix J.

Our second contribution is BEGIN (**Be**nchmarking **G**raph Cont**in**ual Learning), which is a framework for implementation and evaluation of graph CL methods. Evaluation is complicated for graph CL due to additional dependency between instances. Specifically, it is common to utilize instances of one task for other tasks in order to leverage their dependencies, and thus without careful consideration, information that should not be shared among tasks can be leaked among tasks. In order to eliminate potential mistakes in evaluation, BEGIN is **fool-proof** by completely separating the evaluation module from the learning part, where users implement their own graph CL methods. The learning part only has to answer queries provided by the evaluation module after each task is processed. BEGIN is **easy-to-use**, and especially it is easily extended since it is modularized with reusable modules for data processing, algorithm design, training, and evaluation.

Our last contribution is to provide extensive benchmark results of 10 graph CL methods, in terms of 4 evaluation metrics, based on our proposed scenarios and the framework. Especially, to the best of our knowledge, we are the first to apply and evaluate parameter-isolation-based methods to graph CL. Refer to Table 1 for comparison with existing ones. For reproducibility, we provide all source code required for reproducing the benchmark results and documents for users at (Anonymous, 2024).

## 2 RELATED WORKS

**Continual Learning with Independent Data.** Continual learning (CL) methods have been developed mostly for independent data (e.g., images and text), being largely categorized into regularization-, replay-, and parameter-isolation-based methods. Regularization-based methods seek to consolidate knowledge from previous tasks by introducing regularization terms in the loss function. For example, Elastic Weight Consolidation (EWC) (Kirkpatrick et al., 2017) weights to parameters according to the diagonal of the Fisher information matrix, and Memory Aware Synapses (MAS) (Aljundi et al., 2018) computes the importance of parameters according to how sensitive the parameters are. Learning without Forgetting (LwF) (Li & Hoiem, 2017) minimizes the difference between the outputs of a previous model and a new model. Replay-based methods store a sample of data for previous tasks (Chaudhry et al., 2019; Rebuffi et al., 2017). Then, they re-use the data while learning a new task to mitigate forgetting. For example, Gradient Episodic Memory (GEM) (Lopez-Paz & Ranzato, 2017) stores data from previous tasks and prevents the increase of losses on them while learning a new task. Parameter-isolation-based methods learn binary or real-valued masks for weighting parameters or model outputs for each task. For instance, PackNet (Mallya & Lazebnik, 2018) generates binary masks based on the magnitude of learned parameters and re-trains only unmasked ones for new tasks. Piggyback (Mallya et al., 2018) learns real-valued masks and binarizes them to mask parameters for each task. HAT (Serra et al., 2018) learns real-valued masks, which play a role similar to attention modules, for weighting each layer output for each task. There exist a number of frameworks (Lomonaco et al., 2021; Lin et al., 2021) implementing and evaluating CL methods for independent data. However, none of them currently supports CL with graph data.

**Continual Learning with Graph Data (Graph CL).** Due to their expressiveness, graphs are widely used to model various types of data, and thus considerable attention has been paid to machine learning with graph-structured data. Since many such graphs (e.g., online social networks) evolve over time, continual learning is desirable for them. Thus, several graph CL methods have been developed (Febrinanto et al., 2022; Galke et al., 2021; Sun et al., 2023) and are mainly categorized into regularization- (Liu et al., 2021; Cai et al., 2022), replay- (Zhou & Cao, 2021; Wang et al., 2022), and hybrid-based methods (Daruna et al., 2021; Wang et al., 2020a). As an example of replay-based methods, ER-GNN (Zhou & Cao, 2021) samples nodes and uses them for re-training. FGN (Wang et al., 2022) transforms each original node into a form of a feature graph, which takes original features as nodes, and then applies a replay-based method to them (Aljundi et al., 2019). As an example of regularization-based methods, TWP (Liu et al., 2021) stabilizes parameters important in topological aggregation by graph neural networks through regularization. The replay- and regularization-based approaches are combined in CGNN (Wang et al., 2020a). Apart from these primary categories, HPNs (Zhang et al., 2022a) retain and expand abstract knowledge of different levels in the form of prototypes (i.e., embeddings). Given a node, the prototypes relevant to the node are retrieved and used as input for prediction. It is important to note that these approaches have been evaluated under largely inconsistent settings and scenarios, as compared in detail in Appendix J.

Despite these efforts, graph CL is still largely underexplored, especially compared to CL with independent data, and the lack of benchmark frameworks and scenarios is one major reason. To the best of our knowledge, there exist only two benchmarks for graph CL (Carta et al., 2021; Zhang et al., 2022b). However, Carta et al. (2021) supports only graph-level tasks under one incremental setting, and Zhang et al. (2022b) supports node- and graph-level tasks under only two settings. Compared to them, our benchmark and framework are more extensive, as summarized in Table 1.

**Comparison with Other Graph Learning Methods.** Methods have been developed to handle shifts in data distribution between training and test datasets (Li et al., 2022; Wu et al., 2021; Ko et al., 2021; Baek et al., 2020), They, however, differ from CL methods, which typically focus on managing data distributional shifts across different tasks. Moreover, transfer learning (TL) on graph data has been studied for various downstream tasks (e.g., link prediction (Tang et al., 2016)). While CL aims to train a single model for a sequence of tasks, TL aims to adapt a separate model to a new task. In addition, many works on incremental/dynamic graph learning (Rossi et al., 2020; You et al., 2022) focus on the latest snapshot of a dynamic graph to maximize performance on it. However, graph CL aims to preserve performance not only on the current snapshot but also on past ones, which can resemble future snapshots due to seasonality, etc.

## 3 BENCHMARK SCENARIOS

We introduce 31 benchmark scenarios with 4 graph **problems**, 4 incremental **settings**, and 20 real-world **datasets**.

**Common Notations.** We denote each $i$-th task in a sequence by $\mathcal{T}_i$ and use $\mathcal{G} = (\mathcal{V}, \mathcal{E}, \mathcal{X})$ to denote a graph that consists of a set of nodes $\mathcal{V}$, a set of edges $\mathcal{E}$, and node features $\mathcal{X} : \mathcal{V} \to \mathbb{R}^d$, where $d$ is the number of node features. In some of the considered datasets, edge features are given in addition to or instead of node features, and they can be treated similarly to node features. Lastly, we use $\mathcal{Q}$ to indicate the set of queries used for evaluation.

### 3.1 GRAPH LEARNING PROBLEMS OF THREE LEVELS

Our benchmark scenarios are based on various node-, link-, and graph-level problems. Below, we describe node classification, link prediction, and graph classification, as examples.

**Node Classification (NC).** For a node classification (NC) task $\mathcal{T}_i$, the input consists of (a) a graph $\mathcal{G}_i = (\mathcal{V}_i, \mathcal{E}_i, \mathcal{X}_i)$, (b) a labeled set $\mathcal{V}'_i \in \mathcal{V}_i$ of nodes, (c) a set of classes $\mathcal{C}_i$, and (d) the class $f(v) \in \mathcal{C}_i$ for each node $v \in \mathcal{V}'_i$. A query $q$ on a NC task $\mathcal{T}_i$ is a node $v_q \notin \mathcal{V}'_i$ where $f(v_q) \in \mathcal{C}_i$, and its ground-truth answer is $f(v_q)$.

**Link Prediction (LP).** For a link prediction (LP) task $\mathcal{T}_i$, the input consists of a graph $\mathcal{G}_i = (\mathcal{V}_i, \mathcal{E}_i \setminus \mathcal{E}'_i, \mathcal{X}_i)$, where $\mathcal{E}_i$ is the ground-truth set of edges and $\mathcal{E}'_i \in \mathcal{E}_i$ is the set of missing edges among them. A query on an LP task $\mathcal{T}_i$ is a node pair $(u, v) \notin (\mathcal{E}_i \setminus \mathcal{E}'_i)$, and its ground-truth answer is $\mathbf{1}((u, v) \in \mathcal{E}'_i)$, i.e., whether there exists a missing edge between $u$ and $v$ or not.

**Graph Classification (GC).** For a graph classification (GC) task $\mathcal{T}_i$, the input consists of (a) a labeled set of graphs $\mathcal{S}_i$, (b) a set of classes $\mathcal{C}_i$, and (c) the class $f(\mathcal{G}) \in \mathcal{C}_i$ for each graph $\mathcal{G} \in \mathcal{S}_i$. A

query $q$ on a GC task $\mathcal{T}_i$ is a graph $\mathcal{G}_q \notin \mathcal{S}_i$ where $f(\mathcal{G}_q) \in \mathcal{C}_i$, and its ground-truth answer is $f(\mathcal{G}_q)$.

The problem definition of **Link Classification (LC)** problem, which is also used for our benchmark, is extended straightforwardly from that of node classification.

## 3.2 FOUR INCREMENTAL SETTINGS

We introduce four incremental settings for graph CL and describe how they can be applied to the above three problems. We broaden the scope of considerations beyond what was explored in previous studies (refer to Table 1 and Appendix J for in-depth comparisons). When designing them, we aim to decouple changes in different dimensions (tasks, classes, domains, and time) if it does not reflect actual graph dynamics. Moreover, we aim to make the dependency between instances (e.g., connections between nodes) exploitable. Lastly, we make the input for a task (partially) lost in later tasks so that catastrophic forgetting may happen without careful attention to it.

**Task-Incremental (Task-IL).** The set of classes varies with tasks (i.e., $\forall i \neq j, \mathcal{C}_i \neq \mathcal{C}_j$), and they are often disjoint (i.e., $\forall i \neq j, \mathcal{C}_i \cap \mathcal{C}_j = \emptyset$). In addition, for each query in $\mathcal{Q}$, the corresponding task, which we denote by $\mathcal{T}_i$, is provided, and thus its answer is predicted among $\mathcal{C}_i$.

**Class-Incremental (Class-IL).** The set of classes grows over tasks (i.e., $\forall i < j, \mathcal{C}_i \subsetneq \mathcal{C}_j$). In addition, for each query in $\mathcal{Q}$, the corresponding task is **NOT** provided, and thus its answer is predicted among all classes seen so far (i.e., $\bigcup_{j \leq i} \mathcal{C}_j$ for a current task $\mathcal{T}_i$).

**Domain-Incremental (Domain-IL).** We divided entities (i.e., nodes, edges, and graphs) over tasks according to their domains, which are additionally given (see Section 3.3 for real-world examples of domains). Note that, as domains, we use information not directly related to labels, which we aim to infer. The details for each problem are as follows:

- NC & LC: The labeled nodes (or edges) of the input graph are divided into tasks according to their domains. The input graph is fixed (i.e., $\forall i \neq j, \mathcal{G}_i = \mathcal{G}_j$) for all tasks.

- LP: The ground-truth edges are partitioned into (a) the set $\bar{\mathcal{E}}$ of base edges and (b) the set $\tilde{\mathcal{E}}$ of additional edges. The base edges are provided commonly for all tasks, and they are especially useful when answering queries on past tasks. [2] The additional edges $\tilde{\mathcal{E}}$ are provided further for each task according to their domains. For each task $\mathcal{T}_i$, $\bar{\mathcal{E}} \cup \tilde{\mathcal{E}}_i$, where $\tilde{\mathcal{E}}_i$ is the additional edges assigned to $\mathcal{T}_i$, is used as the ground-truth edges (i.e., $\mathcal{E}_i = \bar{\mathcal{E}} \cup \tilde{\mathcal{E}}_i$), and missing ground-truth edges are chosen among $\tilde{\mathcal{E}}_i$, i.e., $\mathcal{E}'_i \subset \tilde{\mathcal{E}}_i$.

- GC: The labeled graphs are divided into GC tasks according to their domains.

**Time-Incremental (Time-IL).** We consider a dynamic graph evolving over time. We denote its $i$-th snapshot by $\mathcal{G}^{(i)} = (\mathcal{V}^{(i)}, \mathcal{E}^{(i)}, \mathcal{X}^{(i)})$. The class set may or may not vary with tasks, and the details of each problem are as follows:

- NC & LC: The input graph for each task $\mathcal{T}_i$ is the $i$-th snapshot $\mathcal{G}^{(i)}$ of the dynamic graph, and labeled nodes (or edges) are given among new nodes (or edges) added to the snapshot (i.e., $\mathcal{V}'_i \subset \mathcal{V}^{(i)} \setminus \mathcal{V}^{(i-1)}$, where $\mathcal{V}^{(0)} = \emptyset$). The label and features of each node (or edge) are fixed over time.

- LP: As in the Domain-IL setting, base edges are used. For each task $\mathcal{T}_i$, the set $\bar{\mathcal{E}}_i$ of base edges so far and the new edges added to the $i$-th snapshot $\mathcal{G}^{(i)}$ of the dynamic graph are used as the ground-truth edges (i.e., $\mathcal{E}_i = \bar{\mathcal{E}}_i \cup (\mathcal{E}^{(i)} \setminus \mathcal{E}^{(i-1)})$, where $\mathcal{E}^{(0)} = \emptyset$). After each task is processed, a subset of $\mathcal{E}^{(i)} \setminus \mathcal{E}^{(i-1)} \setminus \mathcal{E}'_i$ (i.e., new edges that are not used as missing edges) are added as base edges. The features of each node are fixed over time and thus for all tasks.

- GC: The snapshots of the dynamic graph are grouped and assigned to tasks in chronological order. Specifically, for any $i$ and $j$ where $i < j$, every snapshot in the labeled set $\mathcal{S}_i$ of the task $\mathcal{T}_i$ is older than every snapshot in $\mathcal{S}_j$ of $\mathcal{T}_j$. The features of nodes may change over time.

_Remarks on Domain-IL and Time-IL._ While time can be considered as a domain, we deliberately distinguish the Time-IL and Domain-IL settings. Many real-world graphs undergo temporal evolution, establishing an intrinsic connection between time and graph dynamics. However, such a relationship is not apparent for (non-temporal) domains. We assume time-evolving graphs under time-IL settings but not under Domain-IL settings.

_Remarks on Difference from (Zhang et al., 2022b)._ For NC and LC, we fix the input graph for all tasks (i.e., $\forall i \neq j, \mathcal{G}_i = \mathcal{G}_j$) enabling us to study the changes in each dimension (i.e., task, class, and

---

[2] Without base edges, past-task queries need to be answered restrictively using only other-domain edges.

Table 2: **Summary of the considered real-world datasets.**

| Prob. Level | Dataset | # Nodes | # Edges | # Node (Edge) Features | # Classes | Incremental Settings (# Tasks) |
|---|---|---|---|---|---|---|
| | Cora | 2,708 | 10,556 | 1,433 (0) | 7 | Task (3), Class (3) |
| | Citeseer | 3,327 | 9,104 | 3,703 (0) | 6 | Task (3), Class (3) |
| | CoraFull | 19,793 | 126,842 | 8,710 (0) | 70 | Task (35) |
| Node | ogbn-arxiv | 169,343 | 2,332,486 | 128 (0) | 40 | Task (8), Class (8), Time (24) |
| | ogbn-proteins | 132,534 | 79,122,504 | 8 (0) | 2×112 | Domain (8) |
| | ogbn-mag | 736,389 | 10,832,542 | 128 (0) | 349 | Task (128), Class (128), Time (10) |
| | ogbn-products | 2,449,029 | 123,718,024 | 100 (0) | 47 | Class (9) |
| | Twitch | 168,114 | 6,797,557 | 4 (0) | 2 | Domain (21) |
| | Wiki-CS | 11,701 | 431,726 | 300 (0) | 2 | Domain (54) |
| | Bitcoin-OTC | 5,858 | 35,592 | 0 (0) | 21 | Task (3), Class (3), Time (7) |
| Link | ogbl-collab | 235,868 | 1,285,465 | 128 (0) | 2 | Time (50) |
| | Facebook | 134,833 | 1,380,293 | 1 (0) | 2 | Domain (8) |
| | Ask-Ubuntu | 159,313 | 507,988 | 1 (0) | 2 | Time (69) |

| Prob. Level | Dataset | # Graphs (Avg. # Nodes) | Avg. # Edges | # Node (Edge) Features | # Classes | Incremental Settings (# Tasks) |
|---|---|---|---|---|---|---|
| | MNIST | 55,000 (70.6) | 564.5 | 3 (0) | 10 | Task (5), Class (5) |
| | CIFAR10 | 45,000 (117.6) | 941.2 | 5 (0) | 10 | Task (5), Class (5) |
| | Aromaticity | 3,868 (29.7) | 65.4 | 2 (0) | 30 | Task (10), Class (10) |
| Graph | ogbg-molhiv | 41,127 (25.5) | 27.5 | 9 (3) | 2 | Domain (10) |
| | ogbg-ppa | 40,700 (243.1) | 4603.4 | 2 (7) | 37 | Domain (11) |
| | NYC-Taxi | 35,064 (265.0) | 2100.5 | 7 (1) | 2 | Time (16) |
| | Sentiment140 | 5,500 (13.43) | 23.71 | 300 (0) | 2 | Time (11) |

domain) on its own, and we separately consider the dynamics of the input graph in Time-IL settings. It is important to note that the Task-IL and Class-IL settings in (Zhang et al., 2022b) differ from our approach. In their settings, the timing of when edges are added to the input graph is determined solely by the tasks (or classes) that the endpoints belong to. That is, changes in tasks (or classes) and the dynamics of the input graph are perfectly coupled without sufficient justification. This coupling does not reflect actual graph dynamics, which are not solely or predominantly determined by tasks.

### 3.3 REAL-WORLD DATASETS AND BENCHMARK SCENARIOS

We describe 20 real-world datasets and 31 benchmark scenarios based on them under various incremental settings. We summarize the datasets and the scenarios in Table 2.

#### 3.3.1 DATASETS FOR NODE-LEVEL PROBLEMS.

- Cora, Citeseer (Sen et al., 2008), and CoraFull (Bojchevski & Günnemann, 2018) are citation networks. Each node is a scientific publication, and its class is the field of the publication. For Cora and Citeseer, we formulate three binary classification tasks for Task-IL and three tasks with 2, 4, and 6 classes for Class-IL. Similarly, for CoraFull, we formulate 35 binary classification tasks. Note that, one class is left unused in Cora.

- Nodes in ogbn-proteins (Hu et al., 2020; Szklarczyk et al., 2019) are proteins, and edges indicate meaningful associations between proteins. For each protein, 112 binary classes, which indicate the presence of 112 functions, are available. Each protein belongs to one among 8 species, which are used as domains in Domain-IL. Each of the 8 tasks consists of 112 binary classification problems.

- ogbn-arxiv and ogbn-mag (Hu et al., 2020; Wang et al., 2020b) are citation networks, where each node is a research paper. For ogbn-arxiv, its class belongs to 40 subject areas, which are divided into 8 groups for Task-IL, and the number of classes increases by 5 for each task in Class-IL. For ogbn-mag, For Task-IL and Class-IL, among 349 classes indicating fields of studies, we use the 257 classes with at least 10 nodes, and they are divided into 128 groups for Task-IL. For Class-IL, the number of classes grows by 2 in each task. For Time-IL, we formulate 24 and 10 tasks chronologically using publication years for ogbn-arxiv and ogbn-mag, respectively.

- ogbn-products (Hu et al., 2020; Chiang et al., 2019) is a co-purchase network, where each node is a product, and its class belongs to 47 categories, which are divided into 9 groups for Class-IL. The number of classes increases by 5 in each task, and two categories are not used.

- Nodes in Twitch (Rozemberczki & Sarkar, 2021) are users of a video-streaming platform, and edges indicate mutual follower relationships between users. For each user, its class indicates whether the user is joining the affiliate program or not. Each user belongs to one among 21 broadcasting language groups, and they are used as domains for Domain-IL.

#### 3.3.2 DATASET FOR LINK-LEVEL PROBLEMS.

- Bitcoin-OTC (Kumar et al., 2016; 2018) is a who-trust-whom network, where nodes are users of a bitcoin-trading platform. Each directed edge has an integer rating between $-10$ to $10$ and

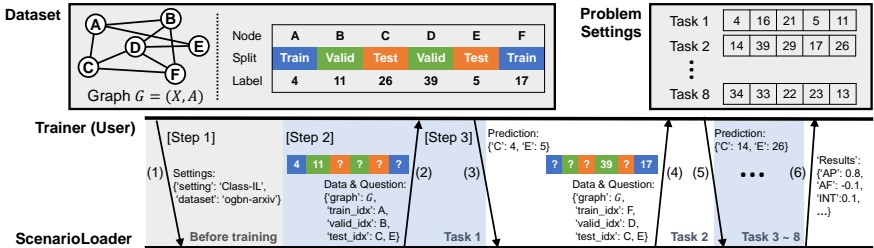

Figure 1: Example communications between the trainer (user code) and the loader of BEGIN.

a timestamp. The ratings are divided into 6 groups. Two of them are used separately for Task-IL and accumulated for Class-IL. For Time-IL, we formulate 7 tasks chronologically using the timestamps, where the signs of the ratings are used as binary classes.

- ogbl-collab (Hu et al., 2020; Wang et al., 2020b) is a co-authorship network between authors with publication years for Time-IL.

- Wiki-CS (Mernyei & Cangea, 2020) is a hyperlink network between computer science articles. Each article has a label indicating one of the 10 subfields it belongs to. The node labels are used as domains, and the edges are divided into 54 groups, according to the labels of their endpoints.

- Nodes in Ask-Ubuntu (Paranjape et al., 2017) are users of an online Q/A platform, and edges indicate interactions between them. The edges are divided into 69 groups chronologically using timestamps for Time-IL.

- In Facebook (Rozemberczki et al., 2019), nodes are pages on Facebook, and edges indicate mutual "likes" between the pages that fall within the same category. Each node is assigned to one of eight categories, and we leverage these categories as domains for Domain-IL.

### 3.3.3 DATASETS FOR GRAPH-LEVEL PROBLEMS.

- Images in MNIST and CIFAR10 (Dwivedi et al., 2020; Achanta et al., 2012) are converted to graphs of super-pixels. There are 10 classes of graphs, and they are partitioned into 5 groups, which are used separately for Task-IL and accumulated for Class-IL.

- Graphs in ogbg-molhiv (Hu et al., 2020; Wu et al., 2018; Landrum et al., 2006) and Aromaticity (Xiong et al., 2019; Wu et al., 2018) are molecules consisting of atoms and their chemical bonds. For ogbg-molhiv, the binary class of each graph indicates whether molecules inhibit HIV replication or not, and we divide the molecules into 20 groups based on structural similarity by the scaffold splitting procedure (Landrum et al., 2006). Aromaticity contains labels representing the number of atoms in each molecule, and in this paper, we divide the molecules into 30 groups based on the labels and formulate Task- and Class-IL settings with 10 tasks.

- Each graph in NYC-Taxi (NYC Taxi & Limousine Commission) shows the amount of taxi traffic between locations in New York City over each hourly period from 2018 to 2021. Specifically, nodes are locations, and there exists a directed edge between two nodes if there existed any taxi customer between them during an hour. The number of such customers is used as the edge weight. The date and time of the corresponding taxi traffic are used to partition the graphs chronologically into 16 groups for Time-IL. The binary class of each graph indicates whether it indicates taxi traffic on weekdays (Mon.-Fri.) or weekends (Sat.-Sun.).

- Graphs in ogbg-ppa (Hu et al., 2020; Szklarczyk et al., 2019; Hug et al., 2016; Zitnik et al., 2019) are protein-protein interactions. We formulate a multi-class (spec., 37-class) classification problem of predicting which taxonomic groups of species each graph comes from. The dataset is preprocessed so that it contains 11 species for each taxonomic group and 100 graphs for each species. We categorize the species so that each category contains exactly one species from each taxonomic group, and the categories are used as domains for Domain-IL.

- Graphs in Sentiment140 (Go et al., 2009) are dependency trees parsed from posts on Twitter. The class of each graph indicates whether the sentiment of the corresponding post is positive or negative. For Time-IL, we formulate 11 tasks chronologically using the timestamps of posts.

## 4 BEGIN: PROPOSED BENCHMARK FRAMEWORK

In this section, we present BEGIN (**Be**nchmarking **G**raph Cont**in**ual Learning), our proposed benchmark framework for making graph continual learning (graph CL) **(a) Easy**: assisting users so that

Figure 2: **(Left) Modularized structure of BEGIN**, our proposed benchmark framework for implementation and evaluation of continual learning methods for graph data. **(Right) An example implementation of EWC with BEGIN.** To implement and benchmark new graph CL methods, users only need to fill out the modularized event functions in the trainer, which then proceeds the training procedure with the event functions.

they can implement new graph CL methods with little effort, **(b) Fool-proof:** preventing potential mistakes of users in evaluation, which is complicated for graph CL, and **(c) Extensive**: supporting various benchmark scenarios, including those described in Section 3. The modularized structure of BEGIN is illustrated on the left side of Figure 2, and below, we focus on its three components.

## 4.1 SCENARIOLOADER (LOADER)

The ScenarioLoader (loader in short) of BEGIN is responsible for communicating with user code (i.e., the training part) to perform a benchmark under a desired incremental setting.

First, the loader receives the entire dataset and the desired incremental setting as inputs. Then, according to the inputs, it processes the dataset into a sequence of tasks, as described in Section 3. Before each task starts, the loader provides (a) the input for the task and (b) the set $\mathcal{Q}$ of queries for evaluation to the user code. Once the user code is done with the current task, the loader receives the predicted answers for the queries in $\mathcal{Q}$. Lastly, if there is no more task to be performed, the loader returns the evaluation results, which are computed by the evaluator module, to the user code. In Figure 1, we provide an example of such communications.

*Remarks on the fool-proofness of* BEGIN. The evaluation part is intentionally concealed from user code, even after all tasks are processed. By hiding the ground-truth answers to the queries, we aim to prevent potential mistakes and misuse by users. It should be noticed that, compared to continual learning with independent data, unintentional information leak is easier to happen for node- and link-level problems due to the dependency between tasks. For example, for an NC task, not only nodes assigned to the current task but also those assigned to previous or future tasks can be used (e.g., for graph convolution) because they are connected by edges. Moreover, our framework additionally restricts information given for each task to prevent potential information leaks. For example, all queries in $\mathcal{Q}$ are asked by the loader and answered by the user code at each evaluation step, even when some of the queries are on unseen tasks. Otherwise, information about the tasks that queries are on can be revealed to the user code and exploited for answering the questions, which is prohibited in the Class-IL setting. Note that the objective of our design is to reduce potential mistakes (i.e., unintentional information leaks), but not to prevent adversarial users from attempting to cheat.

## 4.2 EVALUATOR

BEGIN provides the evaluator to compute basic metrics (spec., accuracy, AUCROC, and Hits@K) based on the ground truth and predicted answers for the queries in $\mathcal{Q}$ provided by the loader. The basic evaluator can easily be extended by users for additional basic metrics. The basic metrics are sent to the loader, and for each basic metric, the basic performance matrix $\mathbf{M} \in \mathbb{R}^{N \times N}$, where $N$ is the number of tasks, is computed. The $(i, j)$-th entry $\mathbf{M}_{i,j}$ indicates the performance on $j$-th task $\mathcal{T}_j$ after the $i$-th task $\mathcal{T}_i$ is processed. Based on $\mathbf{M}$, the following evaluation metrics are computed.

- **Average Performance (AP):** Average performance on $k$ tasks after learning the $k$-th task $\mathcal{T}_k$.

- **Average Forgetting (AF):** Average forgetting on $(k-1)$ tasks after learning the $k$-th task $\mathcal{T}_k$ $(2 \leq k \leq N)$. We measure the forgetting on $\mathcal{T}_i$ by the difference between the performance on $\mathcal{T}_i$ after learning $\mathcal{T}_k$ and the performance on $\mathcal{T}_i$ right after learning $\mathcal{T}_i$.

- **Intransigence (INT)** (Chaudhry et al., 2018) averaged over $k$ tasks: We measure the intransigence on $\mathcal{T}_i$ by the difference between the performances of the Joint model (see Section 5.1) and the target model on $\mathcal{T}_i$ after learning $\mathcal{T}_i$.

- **Forward Transfer (FWT)** (Lopez-Paz & Ranzato, 2017) averaged over $(k-1)$ tasks ($2 \leq k \leq N$): We measure the forward transfer on $\mathcal{T}_i$ by the difference between the performance on $\mathcal{T}_i$ after learning $\mathcal{T}_{i-1}$ and the performance on $\mathcal{T}_i$ without any learning.

Note that **AF** quantifies forgetting of previous tasks, and **INT** measures performance on the current task. **FWT** measures performance on future tasks, to quantify generalizable knowledge retained from previous tasks. Formally, the evaluation metrics are defined as follows:

$$\mathbf{AP} = \sum_{i=1}^{k} \frac{\mathbf{M}_{k,i}}{k}, \ \mathbf{AF} = \sum_{i=1}^{k-1} \frac{\mathbf{M}_{i,i} - \mathbf{M}_{k,i}}{k-1}, \mathbf{INT} = \sum_{i=1}^{k} \frac{\mathbf{M}_{i,i}^{\text{Joint}} - \mathbf{M}_{i,i}}{k}, \mathbf{FWT} = \sum_{i=2}^{k} \frac{\mathbf{M}_{i-1,i} - r_i}{k-1},$$

where $\mathbf{M}^{\text{Joint}}$ is a basic performance matrix of the Joint model, and $r_i$ denotes the performance of a randomly initialized model on $\mathcal{T}_i$.

### 4.3 TRAINER

For usability, BEGIN provides the trainer, which users can extend when implementing new methods. It manages the overall training procedure, including data loading, training, and validation, so that users only have to implement novel parts of their methods. As in (Lomonaco et al., 2021), the trainer divides a training procedure of continual learning as a series of events. For example, the subprocesses in a training procedure contain events called when the trainer (a) receives an input for the current task, (b) trains a model for one iteration for the current task, and (c) handles any necessary pre- and post-processing for the training procedure. Each event is modularized as a function, which users can fill out, and the trainer proceeds with the training procedure with the event functions.

One thing we need to consider is that there can be cases where intermediate results generated in each event must be stored to be used in other events. For example, in the EWC method, a penalty term for preventing catastrophic forgetting should be additionally considered to compute the training loss. To compute the term, the learned parameters and the weights computed from the fisher information matrix on the previous tasks are needed, but they cannot be obtained on the current task. In order to resolve this issue, the trainer provides a dictionary where intermediate results can be stored and shared by events. For the aforementioned EWC method, the learned parameters and computed weights on a task are stored in the dictionary and used for computing the training loss on the following tasks. The right side of Figure 2 shows how the EWC method for node classification can be implemented with BEGIN. In this example, users only need to fill out three event functions: (a) `init_training_states` for initializing the dictionary of training states, (2) `process_train_iteration` for considering the penalty term in the loss function, and (3) `process_after_training` for storing learned weights and compute weights of the parameters for the penalty term. Refer to Appendix G for a further discussion regarding the usability of BEGIN, including provided functionalities and a detailed example implementation.

## 5 BENCHMARK RESULTS

In this section, we provide benchmark results of 10 graph CL methods implemented by BEGIN.

### 5.1 EXPERIMENTAL SETTINGS

For general CL methods, we used LwF (Li & Hoiem, 2017), EWC (Kirkpatrick et al., 2017), MAS (Aljundi et al., 2018), GEM (Lopez-Paz & Ranzato, 2017), PackNet (Mallya & Lazebnik, 2018), Piggyback (Mallya et al., 2018), and HAT (Serra et al., 2018). The parameter-isolation-based methods (i.e., PackNet, Piggyback, and HAT) are applied only to Task-IL since they require knowing which task each query is on. To the best of our knowledge, we are the first to apply parameter-isolation-based methods to graph CL, and we provide their details in Appendix F. For CL methods designed for graph-structured data, we used TWP (Liu et al., 2021), ERGNN (Zhou & Cao, 2021), and CGNN (Wang et al., 2020a). ERGNN and CGNN, which were designed for node-level problems, were not applied to link- and graph-level problems. See Section 2 for brief descriptions of the above methods. For the baseline methods without CL techniques, we used the Bare and Joint models used in (Zhang et al., 2022b). The Bare model follows the incremental learning schemes, but no CL technique is applied to the model. Lastly, the Joint model trains the backbone model directly on the entire dataset, ignoring the CL procure with a sequence of tasks.

We performed all experiments on a Linux server with Quadro RTX 8000 GPUs. Due to the space limitation, we provide the details of the models and training protocol we used in Appendix C.

Table 3: **Results of Average Performance (AP, the higher, the better) and Average Forgetting (AF, the lower, the better)**. In each setting, the best score is in bold, and the second-best score is underlined. O.O.M: out of memory. N/A: methods are not applicable to the problems or scenarios. We report full results in Appendix B.

| Methods | Node Classification (NC) | | | | Link Classification (LC) | | Link Prediction (LP) | | Graph Classification (GC) | | | |
|---|---|---|---|---|---|---|---|---|---|---|---|---|
| | Cora (Task-IL) | Citeseer (Class-IL) | ogbn-proteins (Domain-IL) | ogbn-arxiv (Time-IL) | Bitcoin-OTC (Task-IL) | Bitcoin-OTC (Class-IL) | Wiki-CS (Domain-IL) | ogbl-collab (Time-IL) | CIFAR10 (Task-IL) | MNIST (Class-IL) | ogbg-molhiv (Domain-IL) | NYC-Taxi (Time-IL) |
| Bare | 0.903±0.018 | 0.447±0.040 | 0.690±0.022 | 0.628±0.004 | 0.648±0.071 | 0.243±0.030 | 0.068±0.035 | 0.347±0.025 | 0.646±0.074 | 0.194±0.005 | 0.686±0.039 | 0.709±0.006 |
| LwF | 0.915±0.012 | 0.464±0.039 | 0.714±0.020 | 0.639±0.004 | 0.704±0.032 | 0.242±0.029 | 0.065±0.028 | 0.343±0.028 | 0.840±0.030 | 0.194±0.005 | 0.702±0.048 | 0.711±0.002 |
| EWC | 0.912±0.013 | 0.452±0.037 | 0.761±0.011 | 0.613±0.004 | 0.682±0.046 | 0.242±0.030 | 0.113±0.026 | 0.232±0.014 | 0.784±0.042 | 0.193±0.006 | **0.741±0.032** | **0.720±0.014** |
| MAS | 0.918±0.017 | **0.560±0.024** | 0.694±0.016 | 0.588±0.007 | **0.706±0.033** | 0.244±0.023 | 0.100±0.035 | 0.227±0.075 | 0.762±0.046 | 0.192±0.006 | 0.680±0.037 | 0.714±0.005 |
| GEM | 0.882±0.031 | 0.482±0.033 | **0.810±0.003** | 0.654±0.003 | 0.700±0.035 | **0.287±0.042** | **0.180±0.029** | **0.388±0.063** | 0.769±0.030 | **0.199±0.020** | 0.707±0.033 | 0.706±0.066 |
| TWP | 0.910±0.015 | 0.450±0.037 | O.O.M | 0.593±0.010 | 0.673±0.046 | 0.243±0.028 | 0.131±0.037 | 0.258±0.022 | 0.788±0.044 | 0.190±0.007 | 0.739±0.034 | 0.712±0.003 |
| ERGNN | 0.890±0.031 | 0.457±0.043 | N/A | **0.696±0.004** | N/A | N/A | N/A | N/A | N/A | N/A | N/A | N/A |
| CGNN | 0.911±0.015 | 0.531±0.035 | N/A | 0.676±0.005 | N/A | N/A | N/A | N/A | N/A | N/A | N/A | N/A |
| PackNet | 0.933±0.018 | N/A | N/A | N/A | **0.718±0.030** | N/A | N/A | N/A | **0.855±0.023** | N/A | N/A | N/A |
| Piggyback | **0.938±0.016** | N/A | N/A | N/A | 0.681±0.023 | N/A | N/A | N/A | 0.852±0.021 | N/A | N/A | N/A |
| HAT | 0.920±0.020 | N/A | N/A | N/A | 0.627±0.061 | N/A | N/A | N/A | 0.643±0.078 | N/A | N/A | N/A |
| Joint | 0.924±0.015 | 0.556±0.040 | 0.732±0.002 | 0.734±0.002 | 0.735±0.035 | 0.377±0.031 | 0.229±0.018 | 0.588±0.025 | 0.868±0.021 | 0.900±0.004 | 0.779±0.012 | 0.863±0.004 |

(a) Average Performance (AP)

| Methods | Node Classification (NC) | | | | Link Classification (LC) | | Link Prediction (LP) | | Graph Classification (GC) | | | |
|---|---|---|---|---|---|---|---|---|---|---|---|---|
| | Cora (Task-IL) | Citeseer (Class-IL) | ogbn-proteins (Domain-IL) | ogbn-arxiv (Time-IL) | Bitcoin-OTC (Task-IL) | Bitcoin-OTC (Class-IL) | Wiki-CS (Domain-IL) | ogbl-collab (Time-IL) | CIFAR10 (Task-IL) | MNIST (Class-IL) | ogbg-molhiv (Domain-IL) | NYC-Taxi (Time-IL) |
| Bare | 0.026±0.023 | 0.550±0.066 | 0.131±0.034 | -0.049±0.006 | 0.114±0.087 | 0.722±0.047 | 0.299±0.043 | 0.245±0.022 | 0.270±0.082 | 0.978±0.008 | 0.107±0.053 | 0.028±0.013 |
| LwF | 0.012±0.019 | 0.539±0.060 | 0.054±0.026 | -0.044±0.006 | 0.035±0.021 | 0.726±0.047 | 0.311±0.031 | 0.351±0.033 | 0.030±0.015 | 0.976±0.009 | 0.091±0.057 | **0.014±0.007** |
| EWC | 0.020±0.014 | 0.542±0.065 | 0.074±0.025 | -0.084±0.012 | 0.063±0.046 | 0.724±0.060 | 0.146±0.046 | **0.055±0.037** | 0.055±0.027 | 0.977±0.008 | 0.038±0.028 | 0.071±0.024 |
| MAS | 0.005±0.007 | **0.283±0.050** | 0.012±0.027 | -0.116±0.016 | 0.029±0.025 | 0.726±0.049 | 0.134±0.047 | 0.073±0.066 | 0.077±0.039 | 0.973±0.008 | **0.007±0.031** | 0.056±0.006 |
| GEM | 0.060±0.057 | 0.507±0.053 | **0.003±0.028** | **-0.136±0.006** | 0.026±0.021 | **0.579±0.100** | **0.044±0.057** | 0.299±0.065 | 0.106±0.023 | **0.866±0.080** | 0.065±0.040 | 0.065±0.065 |
| TWP | 0.025±0.018 | 0.545±0.063 | O.O.M | -0.039±0.011 | 0.070±0.059 | 0.721±0.055 | 0.200±0.041 | 0.078±0.017 | 0.056±0.030 | 0.972±0.009 | 0.050±0.034 | 0.055±0.009 |
| ERGNN | 0.052±0.059 | 0.518±0.058 | N/A | -0.095±0.005 | N/A | N/A | N/A | N/A | N/A | N/A | N/A | N/A |
| CGNN | 0.023±0.016 | 0.384±0.055 | N/A | -0.076±0.007 | N/A | N/A | N/A | N/A | N/A | N/A | N/A | N/A |
| PackNet | 0.000±0.000 | N/A | N/A | N/A | **0.000±0.000** | N/A | N/A | N/A | **0.000±0.000** | N/A | N/A | N/A |
| Piggyback | 0.000±0.000 | N/A | N/A | N/A | **0.000±0.000** | N/A | N/A | N/A | **0.000±0.000** | N/A | N/A | N/A |
| HAT | **-0.021±0.019** | N/A | N/A | N/A | 0.134±0.082 | N/A | N/A | N/A | 0.272±0.089 | N/A | N/A | N/A |

(b) Average Forgetting (AF)

## 5.2 AVERAGE PERFORMANCE & AVERAGE FORGETTING

Our benchmark results in terms of final[3] AP and AF are shown in Tables 3a and 3b, respectively.

In Task-IL, PackNet and Piggyback, which are parameter-isolation-based methods, perform best overall in terms of both AP and AF. When we average ranks over all the Task-IL results in the tables, the average ranks of PackNet and Piggyback in terms of AP are 1.3 and 3.0, respectively.[4] Among other methods, the regularization-based methods (i.e., LwF, EWC, MAS, and TWP) tend to perform better than the replay-based ones (i.e., GEM and ERGNN) on NC.

For the node-level problem in Class-IL, most CL methods perform significantly better than the Bare model, and CGNN performs best in terms of both AP and AF. However, for the graph-level problem in Class-IL, most CL methods perform only comparably with the Bare model. In Domain-IL, GEM, which is a replay-based method, performs best overall in terms of AP and second-best in terms of AF among all methods applicable to the setting. In Time-IL, no model outperforms the Bare model consistently on all datasets in terms of AP. That is, it is challenging for current CL methods to deal with temporal dynamics in real-world graphs.

Due to the space limitation, we provide the full results on all scenarios and the detailed analysis in Appendix B with a discussion on the INT and FWT of the CL methods. We also conduct hyperparameter sensitivity analyses and present the results in Appendix D-F.

## 6 CONCLUSION

In this work, based on prior studies, we define four incremental settings for evaluating continual learning methods for graph data (graph CL) based on four dimensions of changes, which are tasks, classes, domains, and time. Then, we apply the settings to node-, link-, and graph-level learning problems. As a result, we provide 31 benchmark scenarios from 20 real-world datasets, which cover all 12 combinations of the 4 incremental settings and the 3 levels of problems. In addition, we propose BEGIN, a fool-proof and easy-to-use benchmark framework for the implementation and evaluation of graph CL methods. Our benchmark results cover more extensive scenarios, CL methods, and evaluation metrics than the existing ones for graph CL, as summarized in Table 1. A limitation of our work is the relatively small number of tasks compared to CL benchmarks in other domains, primarily due to the lack of rich labels in graph data, as further discussed in Appendix H.

---

[3]When $k$ is equal to $N$ in the equations in Section 4.2.

[4]They always perfectly retain knowledge from previous tasks (i.e., AF is 0) since they use parameters disjointly for different tasks.

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

# A ILLUSTRATION AND REAL-WORLD EXAMPLE OF A TIME-EVOLVING GRAPH

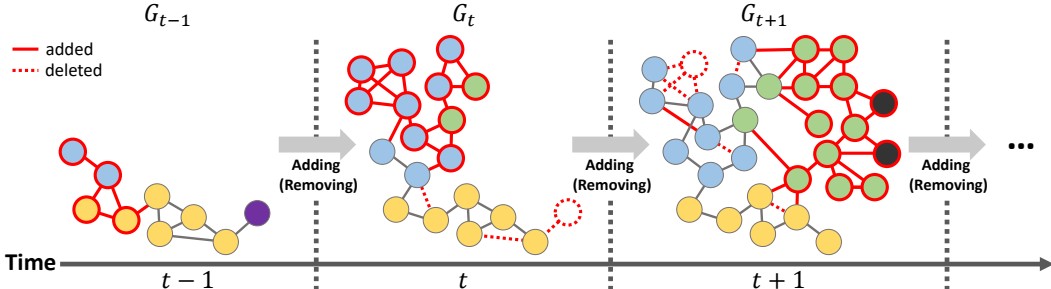

Figure 3: **Illustration of a time-evolving graph.** The colors of nodes represent their classes (or domains). New nodes and edges are depicted using solid red circles and lines, while removed nodes and edges are depicted using dotted red circles and lines. It is important to note that over time, there are changes in (a) the number of nodes, (b) the number of edges, (c) the number of classes (or domains), and (d) the distribution over classes.

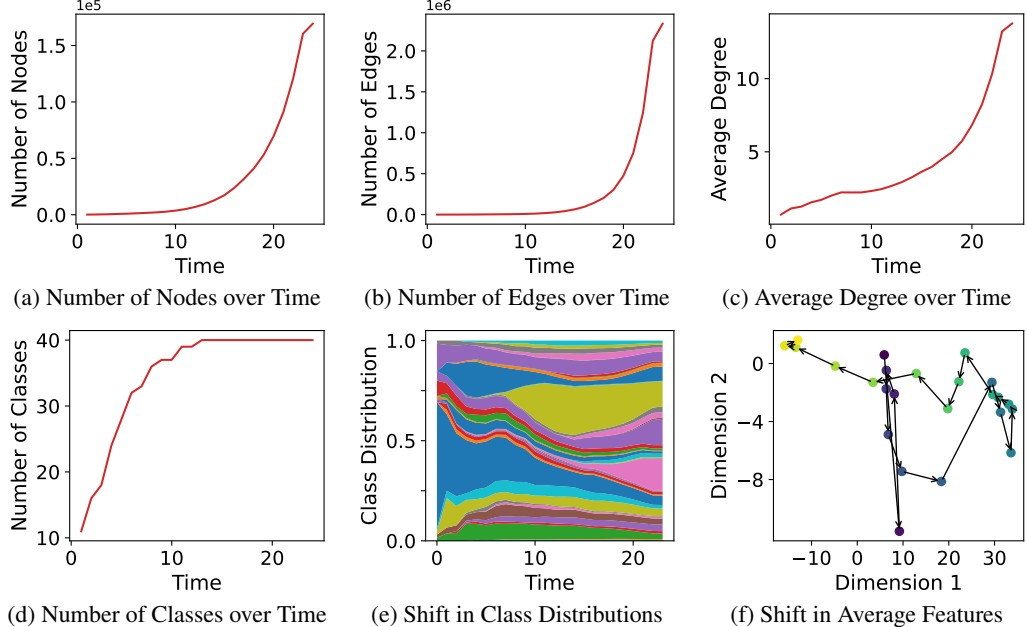

(a) Number of Nodes over Time    (b) Number of Edges over Time    (c) Average Degree over Time

(d) Number of Classes over Time    (e) Shift in Class Distributions    (f) Shift in Average Features

Figure 4: **Changes in six aspects of a real-world time-evolving graph.** In ogbn-arxiv, we observe a gradual increase over time in (a) the number of nodes, (b) the number of edges, (c) the average node degree, and (d) the number of classes. Additionally, the distribution of classes undergoes variations as time progresses, as depicted in (e). Moreover, the average node features exhibit shifts over time, as visualized in (f) using t-SNE (Van der Maaten & Hinton, 2008) to plot the average node features.

Figure 3 provides an example illustration of a time-evolving graph. It demonstrates that both nodes and edges can appear and disappear over time. Additionally, novel classes (or domains) may emerge and subsequently vanish, causing the distribution over classes to shift throughout the timeline.

- At time $t-1$, the graph consists of 9 nodes and 11 edges. There are three classes (or domains) present in the graph: blue, orange, and purple. When observing the graph $G_{t-1}$, we notice that the most prominent class is orange, followed by blue and purple.

- At time $t$, 10 new nodes emerge, accompanied by 15 incident edges. Notably, the node belonging to the purple class disappears, and two existing edges vanish. Additionally, a new class, represented by the color green, emerges in the graph. As a result of these changes, the blue class becomes the most dominant class.

- At time $t + 1$, 10 new nodes belonging to the green class and 2 new nodes belonging to a novel class, represented by the color black, emerge, accompanied by 22 incident edges. Moreover, an existing node and its incident edges are removed. Additionally, a new edge is added among existing nodes, while three existing edges are removed. As a result, the graph now consists of 29 nodes, 40 edges, and 4 classes. Furthermore, the dominance of the blue class diminishes, while the green class emerges as the new dominant class.

Figure 4 illustrates the dynamic changes in six aspects of a real-world graph, ogbn-arxiv, which represents a citation network. The plot shows the changes in the following topological properties: (a) the number of nodes, (b) the number of edges, and (c) the average node degree. Additionally, it presents the variations in (d) the diversity of node classes, (e) the distribution over node classes, and (f) the average node features, visualized using t-SNE (Van der Maaten & Hinton, 2008). In (a)-(c), we observe a progressive increase in the number of available nodes (including train, validation, and test nodes) and edges over time. In (d) and (e), we observe variations in the distribution of node classes over time, accompanied by the emergence of new classes. In (f), we observe that the average node features exhibit shifts over time, as indicated by the directional arrows.

## B  FULL EXPERIMENT RESULTS

### B.1  AVERAGE PERFORMANCE & AVERAGE FORGETTING

In Table 4 and Table 5, we report the results of average performance (AP) and average forgetting (AF) for all 31 benchmark scenarios.

The considered Graph CL methods can be categorized into (a) parameter-isolation-based methods (PackNet, Piggyback, and HAT), (b) replay-based methods (GEM and ERGNN), and regularization-based methods (LwF, EWC, MAS, and TWP). We conducted an analysis to assess the category-based performance differences across various metrics and incremental settings, specifically focusing on Node Classification (NC) problems. To compare these differences, we calculated the rank of each method across all benchmark scenarios, with the exception of the ogbn-mag dataset[5] Subsequently, we visualized the distributions of the ranks for each category using box plots.

As shown in Figure 5, parameter-isolation-based methods outperformed the replay- and regularization-based methods in Task-IL. Furthermore, we observed that replay-based methods generally achieved better overall performance compared to regularization-based methods in all incremental settings, except for Task-IL, in terms of AP and AF. Additionally, by conducting Welch's t-test, we discovered that the category-based performance differences we reported were statistically significant within the 95% confidence interval.

### B.2  INTRANSIGENCE

We report the intransigence (INT) of the considered methods in Table 6. In Task-IL, LwF shows overall the best performance among 10 graph CL methods. Specifically, when we average ranks over all the Task-IL results, LwF takes the first place with an average rank of 2.4, and MAS takes the last place with an average rank of 6.7. Interestingly, the parameter-isolation-based methods, which are best in terms of AP and AF, are not best in terms of INT. The average ranks of PackNet, Piggyback, and HAT are 4.8, 5.9, and 5.1, respectively.

When we average ranks over all the results including all graph problems and all incremental settings without parameter-isolation-based methods, LwF still performs best with an average rank 1.9, and MAS performs worst with an average rank 4.2. However, the difference between regularization-based and replay-based methods was not statistically significant within the 95% confidence interval, as shown in Figure 5.

Interestingly, in terms of INT, many CL methods are outperformed by the Bare model, which focuses only on learning the current task. In other words, the Bare model performs better on the current task than the outperformed CL methods, while it may suffer from catastrophic forgetting for past tasks. This result implies that retaining knowledge from past tasks by the outperformed CL methods is less helpful (to the current task) than using all parameters to learn the current task.

---

[5]Note that GEM ran out of time in ogbn-mag on both Task-IL and Class-IL settings.

Table 4: **Performance in terms of Average Performance (AP, the higher, the better)**. In each setting, the best score is in bold, and the second-best score is underlined. O.O.M: out of memory. O.O.T: out of time (> 24 hours). N/A: methods are not applicable to the problems or scenarios.

| Problem | Node Classification (NC) | | | | | | | | | |
|---|---|---|---|---|---|---|---|---|---|---|
| Methods | Cora (Task-IL) | Citeseer (Task-IL) | ogbn-arxiv (Task-IL) | CoraFull (Task-IL) | ogbn-mag (Task-IL) | Cora (Class-IL) | Citeseer (Class-IL) | ogbn-arxiv (Class-IL) | ogbn-products (Class-IL) | ogbn-mag (Class-IL) |
| Bare | 0.903±0.018 | 0.836±0.029 | 0.650±0.076 | 0.759±0.019 | 0.651±0.016 | 0.541±0.036 | 0.447±0.040 | 0.120±0.004 | 0.105±0.009 | 0.008±0.001 |
| LwF | 0.915±0.012 | **0.859±0.018** | 0.926±0.008 | 0.819±0.017 | O.O.T | 0.550±0.021 | 0.464±0.039 | 0.131±0.009 | 0.106±0.010 | 0.008±0.001 |
| EWC | 0.912±0.013 | 0.837±0.029 | 0.858±0.022 | 0.869±0.022 | 0.887±0.014 | 0.567±0.052 | 0.452±0.037 | 0.123±0.004 | 0.116±0.021 | 0.027±0.004 |
| MAS | 0.918±0.017 | 0.843±0.023 | 0.918±0.008 | 0.972±0.006 | 0.943±0.009 | **0.741±0.014** | **0.560±0.024** | 0.125±0.017 | 0.097±0.018 | 0.008±0.004 |
| GEM | 0.882±0.031 | 0.834±0.029 | 0.906±0.007 | 0.836±0.020 | O.O.T | 0.618±0.029 | 0.482±0.033 | **0.607±0.015** | 0.290±0.042 | O.O.T |
| TWP | 0.910±0.015 | 0.836±0.026 | 0.848±0.015 | 0.900±0.018 | 0.867±0.021 | 0.564±0.031 | 0.450±0.037 | 0.123±0.005 | 0.108±0.014 | 0.027±0.008 |
| ERGNN | 0.890±0.031 | 0.829±0.021 | 0.876±0.015 | 0.900±0.024 | 0.948±0.005 | 0.609±0.026 | 0.457±0.043 | 0.541±0.014 | **0.425±0.078** | **0.170±0.021** |
| CGNN | 0.911±0.015 | 0.829±0.032 | 0.920±0.007 | 0.910±0.016 | 0.953±0.006 | 0.723±0.017 | 0.531±0.035 | 0.477±0.050 | O.O.M | 0.160±0.012 |
| PackNet | 0.933±0.018 | 0.845±0.017 | **0.935±0.007** | 0.973±0.005 | 0.930±0.012 | N/A | N/A | N/A | N/A | N/A |
| Piggyback | **0.938±0.016** | 0.846±0.024 | 0.928±0.007 | **0.977±0.004** | 0.957±0.008 | N/A | N/A | N/A | N/A | N/A |
| HAT | 0.924±0.015 | 0.833±0.043 | 0.905±0.013 | 0.915±0.019 | 0.771±0.022 | N/A | N/A | N/A | N/A | N/A |
| Joint | 0.924±0.015 | 0.851±0.027 | 0.938±0.006 | 0.977±0.005 | 0.969±0.003 | 0.800±0.020 | 0.556±0.040 | 0.654±0.016 | 0.674±0.048 | 0.190±0.006 |

| Problem | Node Classification (NC) | | | | Link Classification (LC) | | | Link Prediction (LP) | | | |
|---|---|---|---|---|---|---|---|---|---|---|---|
| Methods | ogbn-proteins (Domain-IL) | Twitch (Domain-IL) | ogbn-arxiv (Time-IL) | ogbn-mag (Time-IL) | Bitcoin-OTC (Task-IL) | Bitcoin-OTC (Class-IL) | Bitcoin-OTC (Time-IL) | Wiki-CS (Domain-IL) | Facebook (Domain-IL) | ogbl-collab (Time-IL) | Ask-Ubuntu (Time-IL) |
| Bare | 0.690±0.022 | 0.569±0.030 | 0.628±0.004 | 0.296±0.001 | 0.648±0.071 | 0.243±0.030 | 0.696±0.019 | 0.068±0.035 | 0.040±0.009 | 0.347±0.025 | 0.358±0.009 |
| LwF | 0.714±0.020 | 0.587±0.035 | 0.639±0.004 | 0.303±0.001 | 0.704±0.032 | 0.242±0.029 | 0.680±0.021 | 0.065±0.028 | 0.045±0.019 | 0.343±0.028 | 0.358±0.019 |
| EWC | 0.761±0.011 | 0.597±0.016 | 0.613±0.004 | 0.297±0.001 | 0.682±0.046 | 0.242±0.030 | **0.702±0.021** | 0.113±0.026 | 0.042±0.012 | 0.232±0.014 | 0.346±0.011 |
| MAS | 0.694±0.016 | 0.595±0.030 | 0.588±0.007 | 0.286±0.006 | 0.706±0.033 | 0.244±0.023 | 0.687±0.017 | 0.100±0.035 | 0.040±0.014 | 0.227±0.075 | **0.365±0.005** |
| GEM | **0.810±0.003** | **0.602±0.018** | 0.654±0.003 | 0.313±0.002 | 0.700±0.035 | **0.287±0.042** | 0.681±0.019 | **0.180±0.029** | **0.062±0.016** | **0.388±0.063** | 0.360±0.007 |
| TWP | O.O.M | 0.598±0.017 | 0.593±0.010 | 0.295±0.001 | 0.673±0.046 | 0.243±0.028 | 0.664±0.014 | 0.131±0.037 | 0.036±0.013 | 0.258±0.022 | 0.357±0.013 |
| ERGNN | N/A | N/A | **0.696±0.004** | **0.379±0.001** | N/A | N/A | N/A | N/A | N/A | N/A | N/A |
| CGNN | N/A | N/A | 0.676±0.005 | 0.335±0.001 | N/A | N/A | N/A | N/A | N/A | N/A | N/A |
| PackNet | N/A | N/A | N/A | N/A | **0.718±0.030** | N/A | N/A | N/A | N/A | N/A | N/A |
| Piggyback | N/A | N/A | N/A | N/A | 0.681±0.023 | N/A | N/A | N/A | N/A | N/A | N/A |
| HAT | N/A | N/A | N/A | N/A | 0.627±0.061 | N/A | N/A | N/A | N/A | N/A | N/A |
| Joint | 0.732±0.002 | 0.624±0.008 | 0.704±0.005 | 0.382±0.002 | 0.735±0.035 | 0.377±0.031 | 0.800±0.012 | 0.229±0.018 | 0.077±0.011 | 0.588±0.025 | 0.361±0.013 |

| Problem | Graph Classification (GC) | | | | | | | | | |
|---|---|---|---|---|---|---|---|---|---|---|
| Methods | MNIST (Task-IL) | CIFAR10 (Task-IL) | Aromaticity (Task-IL) | MNIST (Class-IL) | CIFAR10 (Class-IL) | Aromaticity (Class-IL) | ogbg-molhiv (Domain-IL) | ogbg-ppa (Domain-IL) | NYC-Taxi (Time-IL) | Sentiment140 (Time-IL) |
| Bare | 0.691±0.081 | 0.646±0.074 | 0.458±0.044 | 0.194±0.005 | 0.175±0.008 | 0.062±0.014 | 0.686±0.039 | 0.299±0.018 | 0.709±0.006 | 0.703±0.010 |
| LwF | 0.965±0.008 | 0.840±0.030 | 0.549±0.044 | 0.194±0.005 | **0.176±0.009** | 0.064±0.013 | 0.702±0.048 | **0.396±0.012** | 0.711±0.002 | 0.702±0.013 |
| EWC | 0.888±0.047 | 0.784±0.042 | **0.588±0.030** | 0.193±0.006 | 0.174±0.009 | 0.064±0.012 | **0.741±0.032** | 0.189±0.026 | **0.720±0.014** | 0.709±0.010 |
| MAS | 0.843±0.044 | 0.762±0.046 | 0.577±0.048 | 0.192±0.006 | 0.175±0.008 | 0.058±0.011 | 0.680±0.030 | 0.145±0.021 | 0.714±0.005 | **0.712±0.008** |
| GEM | 0.894±0.028 | 0.769±0.030 | 0.500±0.045 | **0.199±0.020** | 0.174±0.010 | **0.065±0.013** | 0.707±0.033 | 0.294±0.016 | 0.706±0.066 | 0.674±0.019 |
| TWP | 0.883±0.060 | 0.788±0.044 | 0.564±0.063 | 0.190±0.007 | 0.167±0.012 | 0.062±0.009 | 0.739±0.034 | 0.202±0.021 | 0.712±0.003 | 0.699±0.008 |
| PackNet | **0.972±0.007** | **0.855±0.023** | 0.491±0.089 | N/A | N/A | N/A | N/A | N/A | N/A | N/A |
| Piggyback | 0.962±0.011 | 0.852±0.021 | 0.579±0.042 | N/A | N/A | N/A | N/A | N/A | N/A | N/A |
| HAT | 0.698±0.081 | 0.643±0.078 | 0.455±0.036 | N/A | N/A | N/A | N/A | N/A | N/A | N/A |
| Joint | 0.977±0.007 | 0.868±0.021 | 0.765±0.029 | 0.900±0.004 | 0.521±0.003 | 0.286±0.018 | 0.779±0.012 | 0.801±0.013 | 0.863±0.004 | 0.747±0.010 |

## B.3 FORWARD TRANSFER

We report the forward transfer (FWT) of the considered methods in Table 7. Note that we are able to compute FWT only in Domain-IL. Specifically, in the other settings, the set of classes may vary with tasks (i.e., $\mathcal{C}_i \neq \mathcal{C}_{i-1}$), and thus it may not be possible to infer classes on the next task (i.e., $\mathcal{C}_i$) right after being trained for the current task (i.e., $\mathcal{T}_{i-1}$) without being trained for the next task (i.e., $\mathcal{T}_i$). Except for the ogbn-proteins, where TWP runs out of memory, and the ogbg-ppa datasets, TWP performs best or second-best. On the NC problem, EWC performs best.

Table 5: **Performance in terms of Average Forgetting (AF, the lower, the better)**. In each setting, the best score is in bold, and the second best score is underlined. O.O.M: out of memory. O.O.T: out of time ($> 24$ hours). N/A: methods are not applicable to the problems or scenarios.

| Problem | Node Classification (NC) | | | | | | | | | |
|---|---|---|---|---|---|---|---|---|---|---|
| Methods | Cora (Task-IL) | Citeseer (Task-IL) | ogbn-arxiv (Task-IL) | CoraFull (Task-IL) | ogbn-mag (Task-IL) | Cora (Class-IL) | Citeseer (Class-IL) | ogbn-arxiv (Class-IL) | ogbn-products (Class-IL) | ogbn-mag (Class-IL) |
| Bare | 0.026±0.023 | 0.042±0.030 | 0.325±0.085 | 0.226±0.020 | 0.323±0.017 | 0.565±0.039 | 0.550±0.066 | 0.934±0.008 | 0.850±0.049 | 0.969±0.004 |
| LwF | 0.012±0.019 | 0.011±0.017 | 0.010±0.004 | 0.165±0.015 | O.O.T | 0.534±0.025 | 0.539±0.060 | 0.917±0.010 | 0.845±0.053 | 0.968±0.004 |
| EWC | 0.020±0.014 | 0.041±0.033 | 0.083±0.024 | 0.114±0.024 | 0.078±0.018 | 0.508±0.087 | 0.542±0.065 | 0.924±0.006 | 0.829±0.074 | **0.503±0.068** |
| MAS | 0.005±0.007 | 0.021±0.013 | 0.006±0.004 | 0.001±0.003 | 0.006±0.005 | **0.227±0.039** | **0.283±0.050** | 0.894±0.044 | 0.736±0.071 | 0.574±0.044 |
| GEM | 0.060±0.057 | 0.041±0.026 | 0.036±0.005 | 0.147±0.021 | O.O.T | 0.444±0.046 | 0.507±0.053 | 0.228±0.030 | 0.503±0.084 | O.O.T |
| TWP | 0.025±0.018 | 0.038±0.024 | 0.094±0.013 | 0.082±0.019 | 0.100±0.023 | 0.524±0.041 | 0.545±0.063 | 0.924±0.009 | 0.829±0.061 | 0.655±0.082 |
| ERGNN | 0.052±0.059 | 0.050±0.024 | 0.063±0.012 | 0.078±0.022 | 0.024±0.004 | 0.447±0.045 | 0.518±0.058 | **-0.023±0.071** | **0.375±0.121** | 0.717±0.024 |
| CGNN | 0.023±0.016 | 0.050±0.034 | 0.016±0.003 | 0.065±0.014 | 0.019±0.005 | 0.251±0.034 | 0.384±0.055 | 0.437±0.064 | O.O.M | 0.594±0.011 |
| PackNet | 0.000±0.000 | 0.000±0.000 | **0.000±0.000** | **0.000±0.000** | **0.000±0.000** | N/A | N/A | N/A | N/A | N/A |
| Piggyback | 0.000±0.000 | 0.000±0.000 | **0.000±0.000** | **0.000±0.000** | **0.000±0.000** | N/A | N/A | N/A | N/A | N/A |
| HAT | **-0.021±0.019** | **-0.013±0.018** | 0.032±0.012 | 0.066±0.020 | 0.202±0.022 | N/A | N/A | N/A | N/A | N/A |

| Problem | Node Classification (NC) | | | | Link Classification (LC) | | | Link Prediction (LP) | | | |
|---|---|---|---|---|---|---|---|---|---|---|---|
| Methods | ogbn-proteins (Domain-IL) | Twitch (Domain-IL) | ogbn-arxiv (Time-IL) | ogbn-mag (Time-IL) | Bitcoin-OTC (Task-IL) | Bitcoin-OTC (Class-IL) | Bitcoin-OTC (Time-IL) | Wiki-CS (Domain-IL) | Facebook (Domain-IL) | ogbl-collab (Time-IL) | Ask-Ubuntu (Time-IL) |
| Bare | 0.131±0.034 | 0.055±0.034 | -0.049±0.006 | 0.051±0.001 | 0.114±0.087 | 0.722±0.047 | 0.118±0.027 | 0.299±0.043 | 0.059±0.016 | 0.245±0.022 | **0.045±0.008** |
| LwF | 0.054±0.026 | 0.043±0.036 | -0.044±0.006 | 0.048±0.002 | 0.035±0.021 | 0.726±0.047 | 0.092±0.042 | 0.343±0.028 | 0.059±0.022 | 0.351±0.033 | 0.053±0.015 |
| EWC | 0.074±0.025 | 0.036±0.017 | -0.084±0.012 | 0.044±0.001 | 0.063±0.046 | 0.724±0.060 | 0.099±0.021 | 0.232±0.014 | 0.064±0.014 | 0.055±0.037 | 0.075±0.011 |
| MAS | 0.012±0.027 | 0.039±0.030 | -0.116±0.016 | 0.018±0.008 | 0.029±0.025 | 0.726±0.044 | 0.115±0.025 | 0.227±0.075 | 0.059±0.016 | 0.073±0.066 | 0.063±0.005 |
| GEM | **0.003±0.028** | **0.019±0.018** | **-0.136±0.006** | -0.011±0.003 | 0.026±0.021 | **0.579±0.100** | 0.042±0.033 | 0.388±0.063 | **0.032±0.023** | 0.299±0.065 | 0.063±0.010 |
| TWP | O.O.M | 0.035±0.018 | -0.039±0.011 | 0.044±0.001 | 0.070±0.059 | 0.721±0.055 | 0.193±0.018 | 0.258±0.022 | 0.043±0.015 | 0.078±0.017 | 0.067±0.011 |
| ERGNN | N/A | N/A | -0.095±0.005 | **-0.057±0.001** | N/A | N/A | N/A | N/A | N/A | N/A | N/A |
| CGNN | N/A | N/A | -0.076±0.007 | 0.013±0.001 | N/A | N/A | N/A | N/A | N/A | N/A | N/A |
| PackNet | N/A | N/A | N/A | N/A | **0.000±0.000** | N/A | N/A | N/A | N/A | N/A | N/A |
| Piggyback | N/A | N/A | N/A | N/A | **0.000±0.000** | N/A | N/A | N/A | N/A | N/A | N/A |
| HAT | N/A | N/A | N/A | N/A | 0.134±0.082 | N/A | N/A | N/A | N/A | N/A | N/A |

| Problem | Graph Classification (GC) | | | | | | | | | |
|---|---|---|---|---|---|---|---|---|---|---|
| Methods | MNIST (Task-IL) | CIFAR10 (Task-IL) | Aromaticity (Task-IL) | MNIST (Class-IL) | CIFAR10 (Class-IL) | Aromaticity (Class-IL) | ogbg-molhiv (Domain-IL) | ogbg-ppa (Domain-IL) | NYC-Taxi (Time-IL) | Sentiment140 (Time-IL) |
| Bare | 0.356±0.099 | 0.270±0.082 | 0.182±0.051 | 0.978±0.008 | 0.856±0.021 | 0.694±0.026 | 0.107±0.053 | 0.601±0.019 | 0.028±0.013 | -0.026±0.016 |
| LwF | 0.012±0.004 | 0.030±0.015 | 0.164±0.046 | 0.976±0.009 | 0.857±0.023 | 0.695±0.026 | 0.091±0.057 | 0.493±0.015 | **0.014±0.007** | -0.015±0.011 |
| EWC | 0.084±0.047 | 0.055±0.027 | 0.087±0.030 | 0.977±0.008 | 0.855±0.022 | 0.038±0.028 | **0.062±0.010** | 0.071±0.024 | -0.011±0.011 |  |
| MAS | 0.103±0.048 | 0.077±0.039 | 0.082±0.031 | 0.973±0.008 | 0.849±0.026 | 0.647±0.040 | **0.007±0.031** | 0.074±0.011 | 0.056±0.006 | **-0.030±0.007** |
| GEM | 0.090±0.029 | 0.106±0.023 | 0.169±0.032 | **0.866±0.080** | 0.843±0.022 | **0.337±0.096** | 0.065±0.040 | 0.515±0.031 | 0.065±0.065 | -0.000±0.028 |
| TWP | 0.096±0.062 | 0.056±0.030 | 0.107±0.077 | 0.972±0.009 | **0.842±0.026** | 0.678±0.034 | 0.050±0.034 | 0.074±0.012 | 0.055±0.009 | -0.016±0.010 |
| PackNet | **0.000±0.000** | **0.000±0.000** | **0.000±0.000** | N/A | N/A | N/A | N/A | N/A | N/A | N/A |
| Piggyback | **0.000±0.000** | **0.000±0.000** | **0.000±0.000** | N/A | N/A | N/A | N/A | N/A | N/A | N/A |
| HAT | 0.347±0.099 | 0.272±0.089 | 0.190±0.055 | N/A | N/A | N/A | N/A | N/A | N/A | N/A |

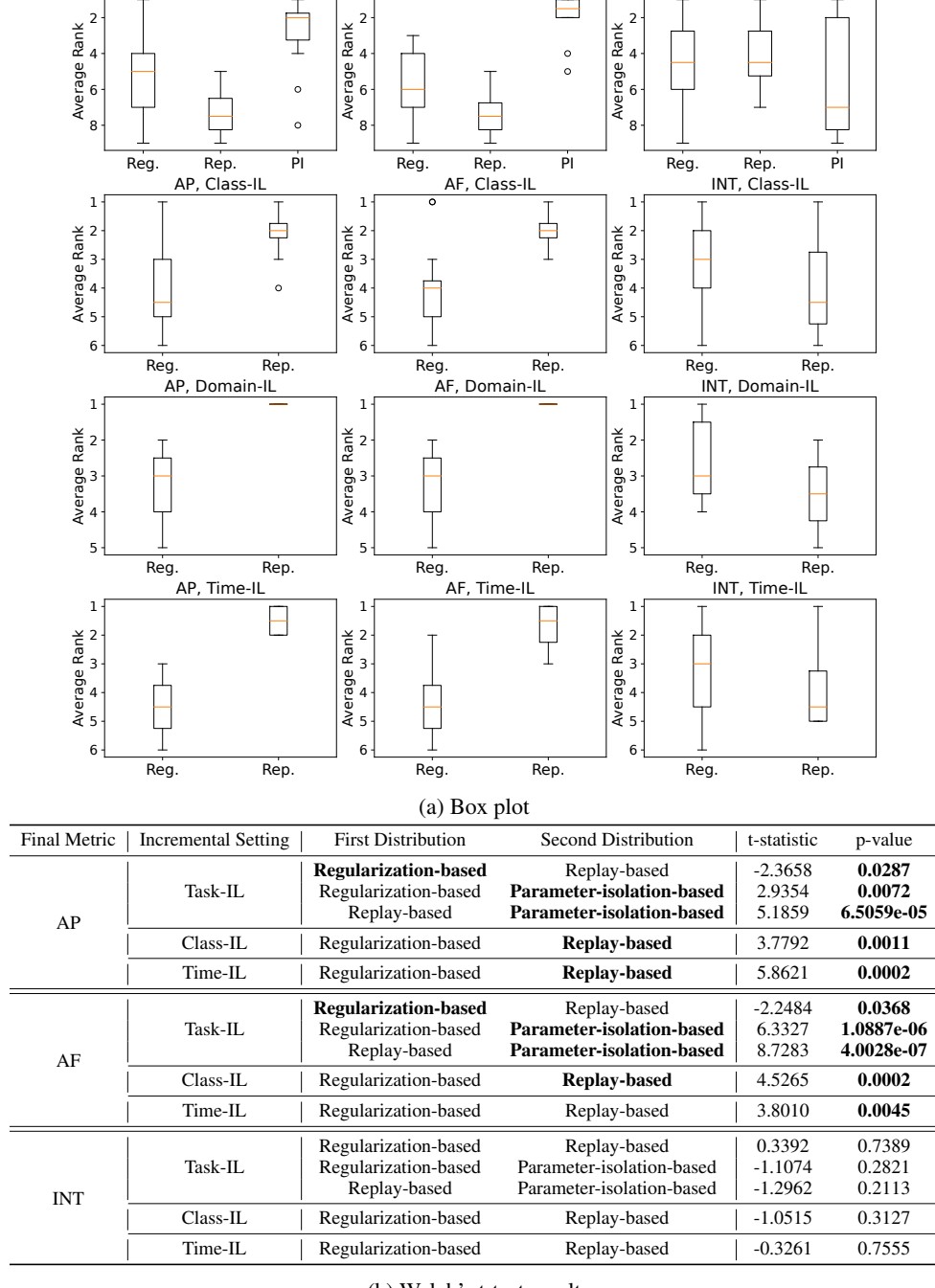

(a) Box plot

| Final Metric | Incremental Setting | First Distribution | Second Distribution | t-statistic | p-value |
|---|---|---|---|---|---|
| AP | Task-IL | **Regularization-based** | Replay-based | -2.3658 | **0.0287** |
| | | Regularization-based | **Parameter-isolation-based** | 2.9354 | **0.0072** |
| | | Replay-based | **Parameter-isolation-based** | 5.1859 | **6.5059e-05** |
| | Class-IL | Regularization-based | **Replay-based** | 3.7792 | **0.0011** |
| | Time-IL | Regularization-based | **Replay-based** | 5.8621 | **0.0002** |
| AF | Task-IL | **Regularization-based** | Replay-based | -2.2484 | **0.0368** |
| | | Regularization-based | **Parameter-isolation-based** | 6.3327 | **1.0887e-06** |
| | | Replay-based | **Parameter-isolation-based** | 8.7283 | **4.0028e-07** |
| | Class-IL | Regularization-based | **Replay-based** | 4.5265 | **0.0002** |
| | Time-IL | Regularization-based | Replay-based | 3.8010 | **0.0045** |
| INT | Task-IL | Regularization-based | Replay-based | 0.3392 | 0.7389 |
| | | Regularization-based | Parameter-isolation-based | -1.1074 | 0.2821 |
| | | Replay-based | Parameter-isolation-based | -1.2962 | 0.2113 |
| | Class-IL | Regularization-based | Replay-based | -1.0515 | 0.3127 |
| | Time-IL | Regularization-based | Replay-based | -0.3261 | 0.7555 |

(b) Welch's t-test result

Figure 5: **Detailed analysis of node classification (NC) performance in terms of AP, AF, and INT.** In the figure, Reg. represents the regularization-based methods (LwF, EWC, MAS, and TWP), Rep. represents the replay-based methods (GEM and ERGNN), and PI represents the parameter-isolation-based methods (PackNet, Piggyback, and HAT). By performing Welch's t-test, we demonstrate that the category-based differences in terms of AP and AF are statistically significant within the 95% confidence interval. We highlighted the $p$-values and outperforming distributions in bold if the difference between the two distributions of ranks is statistically significant.

Table 6: **Performance in terms of Intransigence (INT, the lower, the better)**. In each setting, the best score is in bold, and the second best score is underlined. O.O.M: out of memory. O.O.T: out of time ($> 24$ hours). N/A: methods are not applicable to the problems or scenarios.

| Problem | Node Classification (NC) | | | | | | | | | |
|---|---|---|---|---|---|---|---|---|---|---|
| Methods | Cora (Task-IL) | Citeseer (Task-IL) | ogbn-arxiv (Task-IL) | CoraFull (Task-IL) | ogbn-mag (Task-IL) | Cora (Class-IL) | Citeseer (Class-IL) | ogbn-arxiv (Class-IL) | ogbn-products (Class-IL) | ogbn-mag (Class-IL) |
| Bare | 0.001±0.013 | -0.023±0.052 | 0.004±0.001 | -0.002±0.003 | -0.006±0.001 | **-0.064±0.018** | -0.129±0.053 | **-0.168±0.031** | **-0.132±0.037** | -0.625±0.016 |
| LwF | -0.001±0.005 | **-0.026±0.052** | 0.004±0.002 | -0.002±0.003 | O.O.T | -0.052±0.020 | **-0.139±0.052** | -0.164±0.030 | -0.129±0.040 | -0.624±0.016 |
| EWC | -0.005±0.011 | -0.024±0.054 | 0.008±0.002 | -0.002±0.002 | 0.000±0.003 | -0.052±0.023 | -0.129±0.052 | -0.163±0.031 | -0.124±0.040 | -0.182±0.066 |
| MAS | -0.001±0.010 | -0.016±0.050 | 0.014±0.002 | 0.004±0.004 | 0.016±0.005 | -0.038±0.028 | -0.064±0.050 | -0.142±0.031 | -0.023±0.054 | -0.234±0.049 |
| GEM | -0.001±0.007 | -0.021±0.051 | **0.001±0.001** | -0.002±0.003 | O.O.T | -0.060±0.018 | -0.135±0.053 | -0.037±0.020 | -0.009±0.044 | O.O.T |
| TWP | -0.006±0.009 | -0.022±0.053 | 0.008±0.001 | 0.001±0.002 | 0.000±0.001 | -0.059±0.021 | -0.128±0.053 | -0.162±0.030 | -0.116±0.041 | -0.210±0.058 |
| ERGNN | -0.003±0.011 | -0.022±0.053 | 0.007±0.002 | 0.002±0.003 | **-0.007±0.002** | -0.053±0.018 | -0.118±0.056 | 0.249±0.059 | -0.039±0.045 | -0.538±0.018 |
| CGNN | -0.005±0.009 | -0.022±0.052 | 0.004±0.001 | 0.003±0.003 | **-0.007±0.003** | -0.037±0.011 | -0.102±0.051 | -0.090±0.024 | O.O.M | -0.405±0.011 |
| PackNet | -0.012±0.009 | 0.022±0.023 | 0.003±0.001 | 0.003±0.004 | 0.035±0.010 | N/A | N/A | N/A | N/A | N/A |
| Piggyback | **-0.017±0.012** | 0.020±0.020 | 0.010±0.001 | 0.000±0.003 | 0.018±0.008 | N/A | N/A | N/A | N/A | N/A |
| HAT | 0.015±0.016 | 0.042±0.039 | 0.032±0.012 | **-0.003±0.003** | **-0.007±0.002** | N/A | N/A | N/A | N/A | N/A |

| Problem | Node Classification (NC) | | | | Link Classification (LC) | | | Link Prediction (LP) | | | |
|---|---|---|---|---|---|---|---|---|---|---|---|
| Methods | ogbn-proteins (Domain-IL) | Twitch (Domain-IL) | ogbn-arxiv (Time-IL) | ogbn-mag (Time-IL) | Bitcoin-OTC (Task-IL) | Bitcoin-OTC (Class-IL) | Bitcoin-OTC (Time-IL) | Wiki-CS (Domain-IL) | Facebook (Domain-IL) | ogbl-collab (Time-IL) | Ask-Ubuntu (Time-IL) |
| Bare | -0.067±0.006 | -0.003±0.010 | -0.053±0.013 | -0.003±0.001 | 0.005±0.014 | -0.212±0.067 | 0.003±0.017 | -0.147±0.022 | -0.009±0.015 | 0.031±0.037 | 0.017±0.008 |
| LwF | -0.023±0.021 | -0.009±0.008 | -0.069±0.017 | **-0.008±0.001** | **0.001±0.016** | -0.213±0.064 | 0.041±0.029 | **-0.155±0.034** | -0.015±0.018 | **-0.068±0.046** | 0.008±0.005 |
| EWC | **-0.089±0.009** | -0.013±0.007 | -0.005±0.021 | 0.002±0.001 | 0.004±0.010 | -0.212±0.066 | 0.014±0.021 | -0.042±0.061 | **-0.015±0.027** | 0.333±0.051 | -0.001±0.008 |
| MAS | 0.034±0.026 | **-0.014±0.010** | 0.051±0.020 | 0.036±0.002 | 0.011±0.013 | **-0.215±0.070** | 0.014±0.016 | -0.017±0.031 | -0.010±0.021 | 0.320±0.051 | **-0.008±0.005** |
| GEM | -0.083±0.025 | -0.001±0.009 | 0.004±0.018 | 0.035±0.004 | 0.011±0.013 | -0.160±0.080 | 0.083±0.024 | -0.009±0.041 | -0.008±0.023 | -0.062±0.038 | -0.003±0.004 |
| TWP | O.O.M | -0.012±0.009 | -0.028±0.015 | 0.004±0.001 | 0.009±0.019 | -0.210±0.060 | **-0.030±0.011** | -0.112±0.041 | 0.009±0.018 | 0.284±0.044 | -0.003±0.004 |
| ERGNN | N/A | N/A | **-0.077±0.016** | 0.011±0.001 | N/A | N/A | N/A | N/A | N/A | N/A | N/A |
| CGNN | N/A | N/A | -0.076±0.014 | **-0.008±0.001** | N/A | N/A | N/A | N/A | N/A | N/A | N/A |
| PackNet | N/A | N/A | N/A | N/A | **0.001±0.019** | N/A | N/A | N/A | N/A | N/A | N/A |
| Piggyback | N/A | N/A | N/A | N/A | 0.031±0.033 | N/A | N/A | N/A | N/A | N/A | N/A |
| HAT | N/A | N/A | N/A | N/A | 0.003±0.013 | N/A | N/A | N/A | N/A | N/A | N/A |

| Problem | Graph Classification (GC) | | | | | | | | | |
|---|---|---|---|---|---|---|---|---|---|---|
| Methods | MNIST (Task-IL) | CIFAR10 (Task-IL) | Aromaticity (Task-IL) | MNIST (Class-IL) | CIFAR10 (Class-IL) | Aromaticity (Class-IL) | ogbg-molhiv (Domain-IL) | ogbg-ppa (Domain-IL) | NYC-Taxi (Time-IL) | Sentiment140 (Time-IL) |
| Bare | **0.000±0.003** | 0.004±0.003 | 0.106±0.034 | **-0.045±0.005** | -0.203±0.016 | -0.290±0.040 | -0.004±0.025 | **-0.054±0.013** | 0.098±0.010 | 0.027±0.012 |
| LwF | 0.001±0.001 | **0.002±0.004** | 0.031±0.026 | -0.043±0.006 | **-0.204±0.017** | **-0.293±0.049** | **-0.005±0.019** | -0.053±0.012 | 0.109±0.008 | 0.018±0.012 |
| EWC | 0.021±0.006 | 0.038±0.012 | 0.056±0.036 | -0.042±0.006 | -0.201±0.017 | -0.274±0.050 | 0.007±0.014 | 0.546±0.024 | **0.047±0.018** | **0.008±0.009** |
| MAS | 0.051±0.011 | 0.043±0.013 | 0.063±0.029 | -0.039±0.006 | -0.197±0.018 | -0.243±0.052 | 0.098±0.027 | 0.579±0.016 | 0.067±0.008 | 0.022±0.008 |
| GEM | 0.011±0.005 | 0.013±0.005 | 0.075±0.028 | 0.040±0.052 | -0.191±0.020 | 0.028±0.090 | 0.015±0.019 | 0.029±0.020 | 0.066±0.023 | 0.032±0.016 |
| TWP | 0.016±0.004 | 0.033±0.009 | 0.067±0.024 | -0.036±0.007 | -0.184±0.022 | -0.275±0.052 | -0.003±0.009 | 0.521±0.021 | 0.070±0.010 | 0.022±0.010 |
| PackNet | 0.004±0.003 | 0.012±0.005 | 0.237±0.089 | N/A | N/A | N/A | N/A | N/A | N/A | N/A |
| Piggyback | 0.014±0.007 | 0.019±0.007 | 0.202±0.037 | N/A | N/A | N/A | N/A | N/A | N/A | N/A |
| HAT | **0.000±0.003** | 0.006±0.004 | 0.094±0.028 | N/A | N/A | N/A | N/A | N/A | N/A | N/A |

Table 7: **Performance in terms of Forward Transfer (FWT, the higher, the better)**. In each setting, the best score is in bold, and the second best score is underlined. O.O.M: out of memory.

| Methods | Domain-IL | | | | | |
|---|---|---|---|---|---|---|
| | ogbn-proteins (NC) | Twitch (NC) | Wiki-CS (LP) | Facebook (LP) | ogbg-molhiv (GC) | ogbg-ppa (GC) |
| Bare | 0.159±0.024 | 0.061±0.042 | 0.033±0.018 | 0.012±0.029 | 0.179±0.055 | 0.157±0.013 |
| LwF | 0.164±0.026 | 0.066±0.042 | 0.032±0.016 | **0.019±0.020** | 0.195±0.053 | **0.197±0.015** |
| EWC | **0.171±0.029** | **0.083±0.044** | 0.068±0.022 | 0.009±0.024 | 0.204±0.043 | 0.115±0.017 |
| MAS | 0.165±0.033 | 0.072±0.043 | 0.056±0.027 | 0.006±0.021 | 0.155±0.057 | 0.085±0.014 |
| GEM | 0.166±0.033 | 0.074±0.043 | 0.069±0.028 | 0.008±0.020 | 0.171±0.045 | 0.181±0.020 |
| TWP | O.O.M | 0.075±0.041 | **0.075±0.021** | 0.013±0.023 | **0.214±0.047** | 0.120±0.013 |

## C   DETAILED EXPERIMENTAL SETTINGS

**Models.**  For all experiments, we used GCN (Kipf & Welling, 2017) as the backbone model to compute node embeddings, and the Adam (Kingma & Ba, 2015) optimizer to train the model. For NC, we used a fully-connected layer right after the backbone model to compute the final output. For LC and LP, we additionally used a 3-layer MLP that receives a pair of node embeddings and outputs the final embeddings of the pair. For GC, we used mean pooling to obtain graph embeddings from the output of the backbone model and fed the computed embedding to a 3-layer MLP. For NC and LC, we set the hidden dimension to 256 and used 3 GCN layers. For GC, we set the hidden dimension to 146 and used 4 GCN layers, as in (Dwivedi et al., 2020).

**Training Protocol.**  We set the number of training epochs to $1,000$ for Cora, Citeseer, ogbn-arxiv, and Bitcoin-OTC; 200 for ogbn-proteins and Twitch, Wiki-CS, ogbl-collab, Facebook, and Ask-Ubuntu; and 100 for ogbn-mag, ogbn-products, and all GC datasets. For all datasets except for ogbn-products, we performed full-batch training. For ogbn-products, we trained GNNs with the neighborhood sampler provided by DGL for mini-batch training. For all experiments, we used early stopping. Specifically, for NC and LC, we reduced the learning rate by a factor of 10, if the performance did not improve after 20 epochs, and stopped the experiment if the learning rate became over $1,000$ times smaller than the initial learning rate. For Citeseer, we set the patience to 50 epochs and used the stopping criteria. For LP and GC, we reduced the learning rate by a factor of 10, if performance did not improve after 10 epochs, and stopped the experiment if the learning rate became 100 times ($1,000$ times in Wiki-CS) smaller the initial learning rate.

**Hyperparameter Settings.**  We performed a grid search to find the best hyperparameter settings for the backbone model (e.g., the learning rate and the dropout ratio) for each method. Specifically, we chose the setting where AP on the validation dataset was maximized and reported the results on the test dataset in the selected setting. For evaluation, we conducted ten experiments with different random seeds and averaged the performance over the ten trials. For the CL methods except for TWP, we set the initial learning rate among {1e-3, 5e-3, 1e-2}, the dropout ratio among {0, 0.25, 0.5}, and the weight decay coefficient between {0, 5e-4}. For TWP, we set the initial learning rate between {1e-3, 5e-3}, the dropout ratio among {0, 0.25, 0.5}, and the weight decay coefficient to 0. For the experiments on ogbn-proteins, ogbn-mag (for both task-IL and class-IL settings), ogbn-products, and ogbg-ppa, we fixed the learning rate to 1e-3. For Citeseer, we additionally considered the learning rate 5e-4. For ogbn-mag (for both task-IL and class-IL settings), ogbn-products, and ogbg-ppa, we set the dropout ratio between {0, 0.25}. In addition, for ogbn-products, we restricted the number of maximum neighbors to receive messages to 5, 10, and 10 on the first layer, the second layer, and the third layer, respectively, by using the sampler.

For replay-based methods (e.g., GEM, ERGNN, and CGNN), we set the maximum size of memory the same for a fair comparison. Specifically, we set it to 12 for (a) all experiments on Cora and Citeseer, (b) 210 for CoraFull, (c) $2,000$ for ogbn-arxiv, ogbn-proteins, Twitch, and Ask-Ubuntu, (d) $8,000$ for ogbn-mag, (e) $25,000$ for ogbn-products, (f) 500 for MNIST, CIFAR10, ogbg-molhiv, and ogbg-ppa, (g) 50 for Aromaticity, (h) 800 for NYC-Taxi, (i) $4,000$ for Wiki-CS, (j) $20,000$ for ogbl-collab and Facebook, and (k) 60 for Sentiment140. Following the original papers, we set the margin for quadratic programming to 0.5 for GEM, and we used the Coverage Maximization (CM) sampler and set the distance threshold to 0.5 for ERGNN.

For regularization methods (e.g., LwF, EWC, MAS, TWP, and CGNN), we set the regularization coefficient $\lambda$ to 1.0, 10000.0, 1.0, and 80.0 for LwF, EWC, MAS, and CGNN, respectively. On Cora, Citeseer, ogbn-arxiv, ogbn-proteins, MNIST, CIFAR10, NYC-Taxi, Twitch, Facebook, and Sentiment140, we additionally considered 0.1 for LwF, 100.0 for EWC, and 0.1 for MAS as a potential value of $\lambda$. For TWP, we set beta to 0.01 and $\lambda_l$ to $10,000$, and we chose $\lambda_t$ between $\{100, 1000\}$. Note that CGNN combines regularization- and replay-based approaches, and thus we need to consider both the maximum size of memory and the regularization coefficients.

For parameter-isolation-based methods (e.g. PackNet, Piggyback, and HAT), we set their hyper-parameters as follows. For all parameter-isolation-based methods, we set the weight decay ratio to 0 to maintain the parameters for previous tasks. For PackNet, we set the pruning ratio $p$ to $\exp(\frac{1}{N-1}\log\frac{1}{N})$, where $N$ is the number of tasks, so that the model can learn the parameters of the ratio of $1/N$ at the last task. In addition, we set the number of pre-training epochs to $10\%$ of the

Table 8: **Effects of the number of total tasks on Average Performance, Average Forgetting, and Intransigence.** In each setting, the best score is in bold, and the second best score is underlined. O.O.M: out of memory.

| Methods | ogbn-arxiv (Task-IL) | | | ogbn-arxiv (Class-IL) | | |
|---|---|---|---|---|---|---|
| | $N = 5$ | $N = 8$ | $N = 20$ | $N = 5$ | $N = 8$ | $N = 20$ |
| Bare | 0.649±0.065 | 0.650±0.076 | 0.813±0.035 | 0.185±0.010 | 0.120±0.004 | 0.053±0.007 |
| LwF | 0.895±0.016 | 0.926±0.008 | 0.963±0.010 | 0.199±0.013 | 0.131±0.009 | 0.057±0.009 |
| EWC | 0.794±0.036 | 0.858±0.022 | 0.934±0.009 | 0.188±0.008 | 0.123±0.004 | 0.059±0.012 |
| MAS | 0.882±0.017 | 0.918±0.008 | 0.964±0.009 | 0.187±0.014 | 0.125±0.017 | 0.048±0.007 |
| GEM | 0.873±0.018 | 0.906±0.007 | 0.959±0.009 | **0.626±0.021** | **0.607±0.015** | **0.551±0.011** |
| TWP | 0.792±0.040 | 0.848±0.015 | 0.932±0.017 | 0.189±0.010 | 0.123±0.005 | 0.059±0.010 |
| ERGNN | 0.813±0.022 | 0.876±0.015 | 0.949±0.010 | 0.538±0.023 | 0.541±0.014 | 0.441±0.088 |
| CGNN | 0.894±0.015 | 0.920±0.007 | 0.964±0.009 | 0.533±0.040 | 0.477±0.050 | 0.395±0.046 |
| PackNet | **0.904±0.016** | **0.935±0.007** | **0.971±0.007** | N/A | N/A | N/A |
| Piggyback | 0.893±0.017 | 0.928±0.007 | **0.971±0.007** | N/A | N/A | N/A |
| HAT | 0.874±0.018 | 0.905±0.013 | 0.951±0.014 | N/A | N/A | N/A |
| Joint | 0.906±0.015 | 0.938±0.006 | 0.974±0.007 | 0.681±0.023 | 0.654±0.016 | 0.578±0.016 |

(a) Average Performance (AP, the higher, the better)

| Methods | ogbn-arxiv (Task-IL) | | | ogbn-arxiv (Class-IL) | | |
|---|---|---|---|---|---|---|
| | $N = 5$ | $N = 8$ | $N = 20$ | $N = 5$ | $N = 8$ | $N = 20$ |
| Bare | 0.317±0.074 | 0.325±0.085 | 0.169±0.034 | 0.898±0.017 | 0.934±0.008 | 0.969±0.013 |
| LwF | 0.010±0.004 | 0.010±0.004 | 0.010±0.004 | 0.879±0.016 | 0.917±0.010 | 0.963±0.012 |
| EWC | 0.133±0.037 | 0.083±0.024 | 0.041±0.011 | 0.892±0.018 | 0.924±0.006 | 0.961±0.014 |
| MAS | **0.000±0.001** | 0.006±0.004 | **0.000±0.001** | 0.866±0.016 | 0.894±0.044 | 0.584±0.091 |
| GEM | 0.041±0.008 | 0.036±0.005 | 0.016±0.005 | 0.220±0.033 | 0.228±0.030 | **0.258±0.018** |
| TWP | 0.131±0.040 | 0.094±0.013 | 0.040±0.016 | 0.890±0.021 | 0.924±0.009 | 0.959±0.013 |
| ERGNN | 0.088±0.024 | 0.063±0.012 | 0.021±0.005 | **-0.138±0.086** | **-0.023±0.071** | 0.349±0.095 |
| CGNN | 0.016±0.003 | 0.016±0.003 | 0.011±0.004 | 0.367±0.061 | 0.437±0.064 | 0.511±0.052 |
| PackNet | **0.000±0.000** | **0.000±0.000** | **0.000±0.000** | N/A | N/A | N/A |
| Piggyback | **0.000±0.000** | **0.000±0.000** | **0.000±0.000** | N/A | N/A | N/A |
| HAT | 0.032±0.013 | 0.032±0.012 | 0.023±0.011 | N/A | N/A | N/A |

(b) Average Forgetting (AF, the lower, the better)

| Methods | ogbn-arxiv (Task-IL) | | | ogbn-arxiv (Class-IL) | | |
|---|---|---|---|---|---|---|
| | $N = 5$ | $N = 8$ | $N = 20$ | $N = 5$ | $N = 8$ | $N = 20$ |
| Bare | 0.005±0.001 | 0.004±0.001 | 0.000±0.001 | **-0.127±0.034** | **-0.168±0.031** | **-0.251±0.034** |
| LwF | 0.005±0.001 | 0.004±0.002 | 0.001±0.001 | **-0.127±0.034** | -0.164±0.030 | -0.249±0.034 |
| EWC | 0.005±0.001 | 0.008±0.002 | 0.000±0.002 | -0.126±0.034 | -0.163±0.031 | -0.249±0.034 |
| MAS | 0.025±0.003 | 0.014±0.002 | 0.009±0.002 | -0.105±0.033 | -0.142±0.031 | 0.120±0.072 |
| GEM | 0.001±0.001 | **0.001±0.001** | 0.000±0.002 | -0.026±0.018 | -0.037±0.020 | -0.073±0.027 |
| TWP | 0.009±0.003 | 0.008±0.004 | 0.003±0.002 | -0.125±0.034 | -0.162±0.030 | -0.248±0.035 |
| ERGNN | 0.016±0.004 | 0.007±0.002 | 0.004±0.002 | 0.347±0.073 | 0.249±0.059 | -0.050±0.057 |
| CGNN | **0.000±0.001** | 0.004±0.001 | **-0.001±0.002** | -0.051±0.015 | -0.090±0.024 | -0.158±0.026 |
| PackNet | 0.003±0.001 | 0.003±0.001 | 0.003±0.002 | N/A | N/A | N/A |
| Piggyback | 0.014±0.003 | 0.010±0.001 | 0.003±0.002 | N/A | N/A | N/A |
| HAT | 0.008±0.001 | 0.005±0.002 | 0.001±0.002 | N/A | N/A | N/A |

(c) Intransigence (INT, the lower, the better)

number of training epochs. For Piggyback, we set a threshold $\tau$, which is for determining the binary mask for the parameters of each task, between {1e-1, 1e-2}. For HAT, we set the compressibility $c$ to 0.75 and set the stability $s_{\max}$ to 400.

The hyperparameter settings selected for each considered scenario is available at https://anonymous.4open.science/r/BeGin-C4B0.

## D    EFFECTS OF THE NUMBER OF TOTAL TASKS

We conducted additional experiments to investigate the effect of the number of total tasks $N$ on the performance of graph CL methods. Accordingly, we changed the number of classes considered (additionally) in each task proportionally to the reciprocal of the number of tasks. Under Task- and Class-IL settings on ogbn-arxiv, we measured how the performance changes depending on the number of total tasks from 5 to 20. In Table 8, we report the benchmark final (i.e., when $k$ is equal to $N$) performance in terms of AP, AF, and INT, respectively.

Under Task-IL, most graph CL methods perform better as the number of tasks increases in terms of all 3 metrics, due to the decrease in the number of tasks considered in each task. On the contrary, under Class-IL, the performance tends to degrade in terms of AP and AF as the number of tasks increases since the distribution shift, which each model needs to adapt to, occurs more frequently.

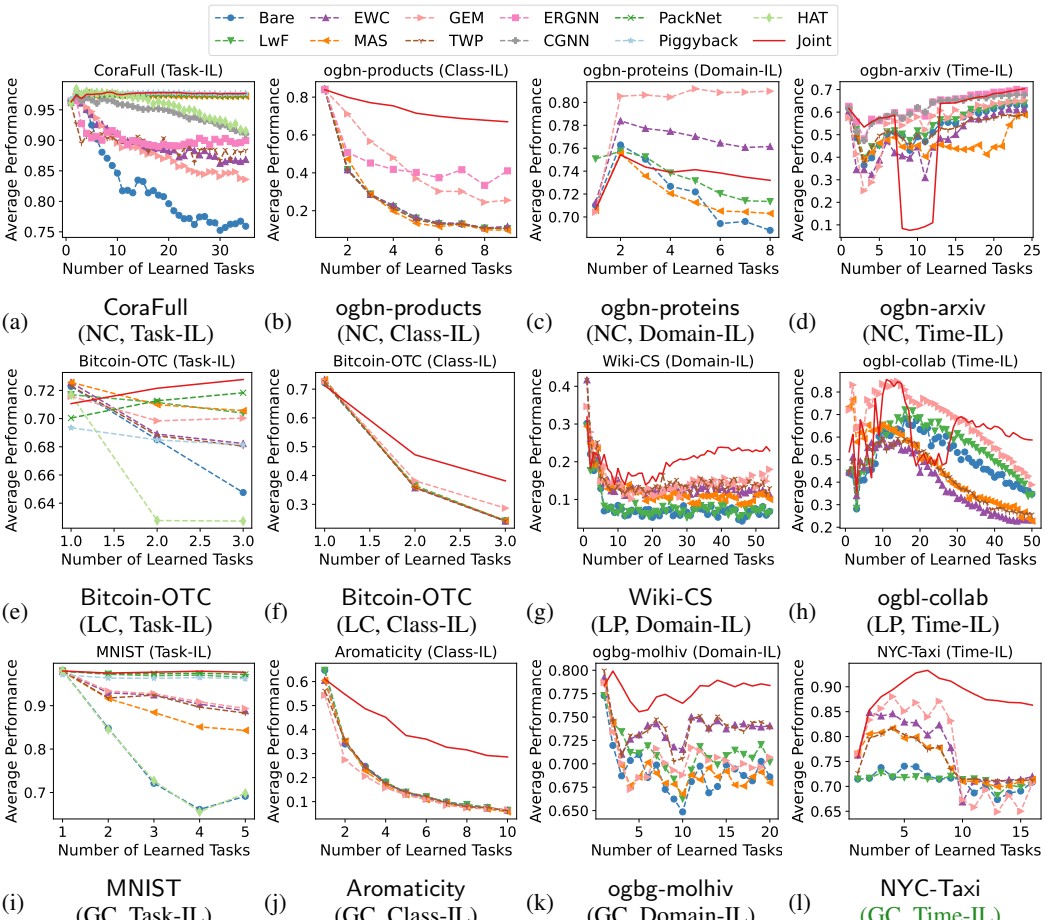

Figure 6: **Change of Average Performance (AP) during continual learning.** Note that the Joint model, which is repeatedly re-trained from scratch, sometimes suffers from instability, especially when training samples for some classes are very limited in Time-IL. The full results are available at https://anonymous.4open.science/r/BeGin-C4B0.

Interestingly, we find that INT tends to change positively as the number of tasks increases because INT does not consider the forgetting of previous tasks.

# E PERFORMANCE CURVE (EFFECTS OF THE NUMBER OF LEARNED TASKS)

In Figure 6, we report the performance curves, which show how the average performance (AP) (when $k$ is equal to $n$) changes depending on the number of learned tasks $n$. Note that most considered methods, including the Joint model, tend to perform worse, as the number of learned tasks increases, since they need to retain more previous knowledge. However, opposite trends are observed on ogbn-arxiv, ogbg-molhiv, and Sentiment140, which we guess have slighter or easier-to-adapt distribution shifts. Also, on Bitcoin-OTC under Task-IL, the performance of the Joint model and that of graph CL methods show different tendencies. We provide the performance curves under all considered scenarios in the online appendix, which is available at **https://anonymous.4open.science/r/BeGin-C4B0**.

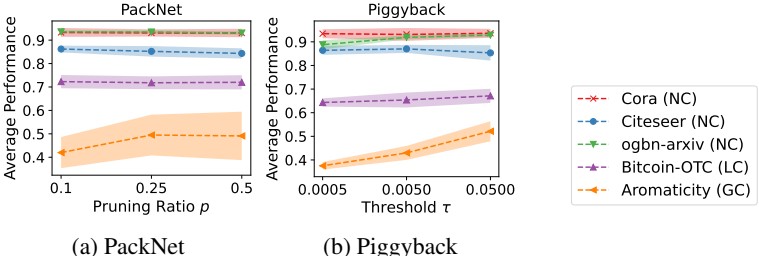

Figure 7: **Effect of hyperparameters of PackNet and Piggyback on Average Performance (AP).** We represent the changes in average performances with dotted lines, and the colored areas represent their standard deviations.

# F    PARAMETER-ISOLATION-BASED METHODS

To the best of our knowledge, we are the first to apply parameter-isolation-based methods (spec., PackNet, Piggyback, and HAT), which have been used for independent data (e.g., images and text), to graph-related problems (spec., NC, LC, and GC). The methods learn binary or real-valued masks for weighting parameters or model outputs for each task. Note that, since they require knowing which task each query is on, they are applicable only to Task-IL. Below, we describe how we apply the methods to graph CL.

**PackNet (Mallya & Lazebnik, 2018).** learns binary masks for network parameters as follows:

1. For the current task, pre-train all parameters that have been unmasked so far.

2. Mask the set of pre-trained parameters whose magnitude is in top $(1-p) \times 100\%$, where $p$ is the pruning ratio.

3. Re-initialize the unmasked ones to $0$ and re-train only the newly masked parameters for the current task.

4. After the re-training, fix the masked parameters to their current values to prevent forgetting

5. Repeat the above processes for the next task.

In our setting, we applied binary masks to both GCN (the backbone model) and fully-connected layer(s).

**Piggyback (Mallya et al., 2018).** requires a pre-trained network, which is fixed over tasks, and learns real-valued masks for each task. For all node-level problems, we used Deep Graph Infomax (Velickovic et al., 2019) to pre-train the GCN layers. For all link-level problems, we applied Deep Graph Infomax to pre-train the GCN layers and the 3-layer MLP. For the graph-level problems, we used InfoGraph (Sun et al., 2019) to pre-train the GCN layers and the 3-layer MLP. Note that any unsupervised training method for graphs can be used instead for pre-training.

In the training phase, Piggyback learns real-valued masks, which are multiplied with the pre-trained parameters, end-to-end for each task. In the test phase, it uses a binarizer to transform the mask entries above the threshold hyperparameter $\tau$ into $1$ and the others into $0$. Lastly, for each task, it applies the binarized masks for the task to the pre-trained network to obtain the network to be used for task. As in PackNet, we applied the above steps to both GCN and fully-connected layer(s).

**HAT (Serra et al., 2018).** learns real-valued masks, which play a role similar to attention modules, for the outputs of each layer, while the above two methods apply binary masks to network parameters. Specifically, we learn real-valued masks for each layer, and perform element-wise multiplication of them and the outputs of each layer. When we implemented HAT for graph CL, we applied all the techniques introduced in Section 2 of the original paper. For each task, we applied the above steps to GCN and fully-connected layer(s).

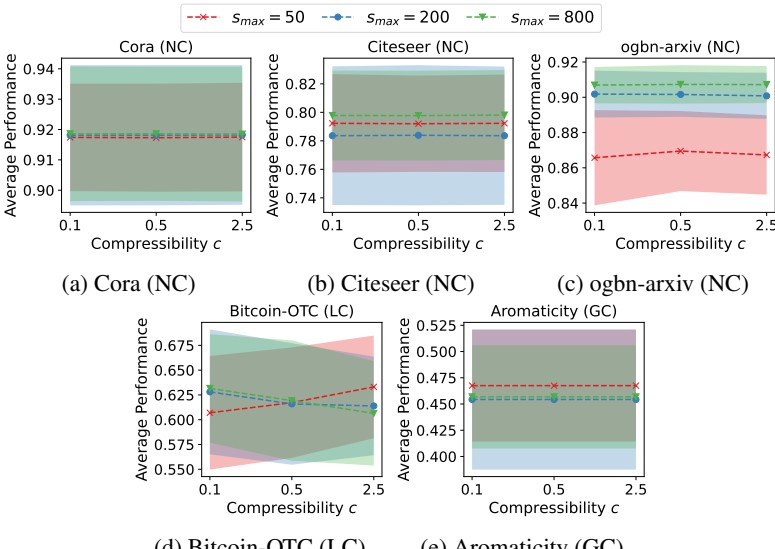

Figure 8: **Effect of hyperparameters of HAT on Average Performance (AP).** The dotted lines represent the means, and the colored regions represent the standard deviations.

**Effects of Hyperparameters.** We performed extra experiments to investigate the effects of their hyperparameters on the performance of the parameter-isolation-based methods. For PackNet, we examined the effect of the pruning ratio $p$ while varying it among $\{0.1, 0.25, 0.5\}$. For piggyback, we examined the effect of the threshold $\tau$ while varying it among $\{5e\text{-}2, 5e\text{-}3, 5e\text{-}4\}$. For HAT, we checked the effect of the stability parameter $s_{max}$ and the compressibility parameter $c$ while varying them among $\{50, 200, 800\}$ and $\{0.1, 0.5, 2.5\}$, respectively.

In Figures 7-8, we report the AP scores for NC on Cora, Citeseer, and ogbn-arxiv; LC on Bitcoin-OTC, and GC on Aromaticity. In Figure 7, we show the effect of the hyperparameters of PackNet and Piggyback. In general, PackNet was not sensitive to the choice of pruning ratio $p$. Specifically, the effect of the changes in the pruning ratio was significant only on Aromaticity, where increasing the pruning ratio from $0.1$ to $0.25$ greatly increased the accuracy. Regarding Piggyback, its accuracy changed gradually with respect to the threshold $\tau$, and the tendencies vary with datasets. In Figure 8, we report the results regarding HAT. Its accuracy depended heavily on the stability $s_{max}$, especially on Citeseer, ogbn-arxiv, and Aromaticity, while HAT was relatively insensitive to the choice of the compressibility parameter $c$. The compressibility parameter $c$ significantly affected the accuracy of HAT only on Bitcoin-OTC.

## G USABILITY OF BEGIN

As shown in Figure 2, BEGIN is easy-to-use, thanks to its built-in training and evaluation procedures and modularized structure. In this section, we will provide a more detailed explanation of the usability of BEGIN.

### G.1 SUPPORTED EVENT FUNCTIONS

If a user-defined event function is not provided, the trainer performs training and evaluation with the corresponding basic pre-implemented operations. Thus, users do not need to implement the whole training and evaluation procedure, but only the necessary parts. Currently, users can override the following built-in event functions:

- `initTrainingStates()`: This function is called only once, when the training procedure begins.

- `prepareLoader()`: This function is called once for each task when generating dataloaders for training, validation, and test. Given the dataset for each task, it should return dataloaders for training, validation, and test.

- `processBeforeTraining()`: This function is called once for each task, right after the `prepareLoader()` event function terminates.

- `processTrainIteration()`: This function is called for every training iteration. When the current batched inputs, model, and optimizer are given, it should perform a single training iteration and return the information or outcome during the iteration.

- `processEvalIteration()`: This function is called for every evaluation iteration. When the current batched inputs and trained model are given, it should perform a single evaluation iteration and return the information or outcome during the iteration.

- `inference()`: This function is called for every inference step in the training procedure.

- `beforeInference()`: This function is called right before `inference()` begins.

- `afterInference()`: This function is called right after `inference()` terminates.

- `_reduceTrainingStats()`: This function is called at the end of every training step. Given the returned values of the `processTrainIteration()` event function, it should return overall and reduced statistics of the current training step.

- `_reduceEvalStats()`: This function is called at the end of every evaluation step. Given the returned values of the `processEvalIteration()` event function, it should return overall and reduced statistics of the current evaluation step.

- `processTrainingLogs()`: This function is called right after the `reduceTrainingStats()` event function terminates.

- `processAfterEachIteration()`: This function is called at the end of the training iteration. When the outcome from `reduceTrainingStats()` and `reduceEvalStats()` are given, it should determine whether the trainer should stop training for the current task or not.

- `processAfterTraining()`: This function is called once for each task when the trainer completes training for the current task.

The arguments and detailed roles of these event functions are provided in the official document of BEGIN, which is available at https://anonymous.4open.science/r/BeGin-C4B0.

### G.2 DETAILED EXAMPLE: ELASTIC WEIGHT CONSOLIDATION (EWC)

In this section, we provide a step-by-step example of implementing EWC (Kirkpatrick et al., 2017) for node classification using BEGIN. This example is the extended version of the one in Figure 2. EWC uses the weighted L2 penalty term based on the learned weights and fisher information matrices from the previous tasks. The overall loss function $\mathcal{L}(\cdot)$ for EWC can be formulated as follows:

$$\mathcal{L}(\Theta) = \mathcal{L}_i(\Theta) + \sum_{j=1}^{i-1} \frac{\lambda}{2} F_j(\Theta - \Theta_j^*)^2, \tag{1}$$

where $\mathcal{L}$ is the loss function, $\Theta$ is the parameter of the model, $\mathcal{L}_i$ is the loss for the current task $i$, $\lambda$ is the hyperparameter for the L2 penalty term, $\Theta_j^*$ is the learned weights at task $j$, and $F_j$ is the fisher information matrix calculated from $\Theta_j^*$.

**Step 1. Extending the base trainer** For each problem considered in this paper, BEGIN provides a *base* trainer class that makes the training behavior exactly the same as the Bare model, described in Section 5.1. Based on the base class, users can implement their CL algorithm by extending the class and substituting some default event functions with user-defined ones.

```
for begin.trainers.nodes import NCTrainer
class NCClassILEWCTrainer(NCTrainer):
    pass
```

**Step 2. Setting initial states for the algorithm**   As stated in Eq. equation 1, EWC requires storing the learned weights and Fisher information matrices from the previous tasks to compute the regularization term. However, they cannot be obtained on the current task. In order to resolve this issue, the trainer provides a dictionary called `training_states`. The dictionary can be used to store intermediate results and can be shared by events in the form of an argument (i.e., input parameter) of the event functions. To set the initial states, the user can extend the base trainer with their modified `initTrainingStates()` event function, which initializes the states for running EWC.

```
for begin.trainers.nodes import NCTrainer
class NCClassILEWCTrainer(NCTrainer):
    def initTrainingStates(self, model, optimizer):
        return {'fishers': [], 'params': []}
```

**Step 3. Computing and Storing Fisher information matrix**   In order to compute the penalty term at task $i$, we need the learned weights $\Theta_j^*$ and Fisher information matrix $F_j$ for every task $j < i$. Hence, we need to store them at the end of each task, and this can naturally be implemented in the event function `processAfterTraining()`, which is called at the end of each task. In the example below, `curr_training_states['params'][j-1]` and `curr_training_states['fishers'][j-1]` store the learned weights and the Fisher information matrix of task $j$, respectively.

```
class NCClassILEWCTrainer(NCTrainer):
    def initTrainingStates(self, model, optimizer):
        return {'fishers': [], 'params': []}

    def processAfterTraining(self, task_id, curr_dataset, curr_model,
        curr_optimizer, curr_training_states):
        super().processAfterTraining(task_id, curr_dataset, curr_model,
            curr_optimizer, curr_training_states)
        params = {name: torch.zeros_like(p) for name, p in
            curr_model.named_parameters()}
        fishers = {name: torch.zeros_like(p) for name, p in
            curr_model.named_parameters()}
        train_loader = self.prepareLoader(curr_dataset,
            curr_training_states)[0]

        total_num_items = 0
        for i, _curr_batch in enumerate(iter(train_loader)):
            curr_model.zero_grad()
            curr_results = self.inference(curr_model, _curr_batch,
                curr_training_states)
            curr_results['loss'].backward()
            curr_num_items =_curr_batch[1].shape[0]
            total_num_items += curr_num_items
            for name, p in curr_model.named_parameters():
                params[name] = p.data.clone().detach()
                fishers[name] += (p.grad.data.clone().detach() ** 2) *
                    curr_num_items

        for name, p in curr_model.named_parameters():
            fishers[name] /= total_num_items

        curr_training_states['fishers'].append(fishers)
        curr_training_states['params'].append(params)
```

**Step 4. Computing $\mathcal{L}(\Theta)$ for regularization**   The penalty term in Eq. equation 1 is used for regularization during a backpropagation process. The computation of the term should be performed at the end of training for every task, and thus it is implemented in `afterInference()`. In

the event function, the argument (i.e., input parameter) `curr_training_states` contains the Fisher information matrices and the previously learned weights based on which the penalty term `loss_reg` is computed. The event function also has the argument `results`, which contains the prediction result and loss of the current model computed in the `inference()` function. Thus, the overall loss including the penalty term can be obtained by summing up `results['loss']` and `loss_reg`.

```python
class NCClassILEWCTrainer(NCTrainer):
    def initTrainingStates(self, model, optimizer):
        return {'fishers': [], 'params': []}

    def processAfterTraining(self, task_id, curr_dataset, curr_model,
        curr_optimizer, curr_training_states):
        super().processAfterTraining(task_id, curr_dataset, curr_model,
            curr_optimizer, curr_training_states)
        params = {name: torch.zeros_like(p) for name, p in
            curr_model.named_parameters()}
        fishers = {name: torch.zeros_like(p) for name, p in
            curr_model.named_parameters()}
        train_loader = self.prepareLoader(curr_dataset,
            curr_training_states)[0]

        total_num_items = 0
        for i, _curr_batch in enumerate(iter(train_loader)):
            curr_model.zero_grad()
            curr_results = self.inference(curr_model, _curr_batch,
                curr_training_states)
            curr_results['loss'].backward()
            curr_num_items =_curr_batch[1].shape[0]
            total_num_items += curr_num_items
            for name, p in curr_model.named_parameters():
                params[name] = p.data.clone().detach()
                fishers[name] += (p.grad.data.clone().detach() ** 2) *
                    curr_num_items

        for name, p in curr_model.named_parameters():
            fishers[name] /= total_num_items

        curr_training_states['fishers'].append(fishers)
        curr_training_states['params'].append(params)

    def afterInference(self, results, model, optimizer, _curr_batch,
        training_states):
        loss_reg = 0
        for _param, _fisher in zip(training_states['params'],
            training_states['fishers']):
            for name, p in model.named_parameters():
                l = self.lamb * _fisher[name]
                l = l * ((p - _param[name]) ** 2)
                loss_reg = loss_reg + l.sum()
        total_loss = results['loss'] + loss_reg
        total_loss.backward()
        optimizer.step()
        return {'loss': total_loss.item(),
                'acc': self.eval_fn(results['preds'].argmax(-1),
                    _curr_batch[0].ndata['label'][_curr_batch[1]].to(self.device))}
```

The above code presents the complete implementation of EWC for node classification under Class-IL. Similarly, various CL algorithms can be developed by modifying specific event functions, without the need to manage the overall training and evaluation procedures.

Table 9: **Lines of Python code to implement each algorithm (the lower, the better).** On average, with BEGIN, we can implement the same graph CL method with about $28.6\%$ fewer lines of code. For ERGNN, we only considered the code for node classification with the CM sampler.

| Method | LwF | EWC | MAS | GEM | TWP | ERGNN | Average |
|---|---|---|---|---|---|---|---|
| **BEGIN (ours)** | **123** | **170** | **156** | **254** | **284** | **124** | **181.8** |
| CGLB (Zhang et al., 2022b) | 171 | 236 | 282 | 277 | 293 | 144 | 233.8 |

## G.3 Lines of Code (LoC) Comparision

In Table 9, we report the lines of Python code for the CL methods commonly implemented in BEGIN and CGLB (Zhang et al., 2022b).[6] For each CL method, we calculated the sum of the lines of the Python code for node classification and graph classification (if available) in Task-IL and Class-IL settings. When counting the number of lines, we ignored comments, empty lines, and unnecessary imports in the codes. It should be noticed that, with BEGIN, we can implement the same graph CL method with about $28.6\%$ fewer lines of code, on average.

## H Limitations of Our Work

In our benchmark scenarios, the number of tasks is relatively small compared to CL benchmarks for independent data (e.g., images). This is because real-world graph datasets have a limited number of classes, with a maximum of 257 in the datasets we considered. In contrast, image datasets like ILSVRC2012 often have 1000 or more classes (Lopez-Paz & Ranzato, 2017; Rebuffi et al., 2017). Moreover, it is important to highlight that increasing the number of tasks for real-world graph datasets is not trivial without artificial manipulation of the datasets. Due to this difficulty, the number of tasks considered in previous studies of graph continual learning (ERGNN, CGNN, and CGLB) are only 41, 9, and 70, respectively. In the future, we plan to update our framework and benchmarks as datasets with richer labels become available, enabling a broader exploration of graph continual learning.

## I Detailed Information about Benchmark Scenarios and Assets

### I.1 Detailed Description of Benchmark Scenarios

In this section, we provide detailed information of the considered benchmark scenarios. We provide a summary of all the benchmark scenarios in Table 10.

**Common Details.** In all settings except for Time-IL, we randomized the sequence of CL tasks for each run. For task-IL and class-IL settings, we randomly grouped the classes to configure the designated number of tasks, ignoring the remaining classes.[7] We chose the design since there is no available information on the order of labels or domains and also to reduce the effect of the order of CL tasks on performance and benchmark results. To ensure a fair comparison, we conducted ten runs using fixed random seeds (spec., 0, 1000, 2000, $\cdots$, 9000) for all experiments.

**Dataset-specific Details.** In ogbn-products, we ignored the 47-th class since there is only one node with the class. For Task-IL and Class-IL settings in Bitcoin-OTC, we formed 6 groups of links based on their weights: $[-10, -9), [-9, 0), [3, 4), [4, 5), [5, 6)$, and $[6, 10]$. This grouping was implemented to mitigate class imbalance. In Aromaticity, we only considered the 30 classes with at least 20 graphs, as in (Zhang et al., 2022b), for the same reason. In Wiki-CS, there are $\binom{10}{2} + \binom{10}{1} = 55$ different combinations of the classes (or domains) of links' endpoints, and we ignored one combination since there is no link with that specific combination.

For NYC-Taxi, we first split 265 locations into seven groups based on borough information provided by (NYC Taxi & Limousine Commission) and employed a one-hot indicator vector as node features. To generate graphs, for every hourly period and for every ordered pair $(A, B)$ of the locations, we counted the number of taxi customers departed from location $A$ and bounded to location $B$. Then, if and only if the count is nonzero, we made a direct edge from $A$ to $B$, using the count as the feature

---

[6]We used the final version of CGLB on Jul 30, 2023 (commit `f514016`).

[7]For example, one class is left unused in Cora, since we used 6 out of 7 classes in the dataset.

Table 10: **Summary of the considered benchmark scenarios**

| Problem | Dataset | Incremental Setting | # Tasks | Task Split Type | Train/Val/Test Split Type | Performance Metric |
|---|---|---|---|---|---|---|
| Node Classification (NC) | Cora | Task-IL, Class-IL | 3 | Class label (2 classes per task) | Original Split | Accuracy |
| | Citeseer | Task-IL, Class-IL | 3 | Class label (2 classes per task) | Original Split | |
| | ogbn-arxiv | Task-IL, Class-IL | 8 | Class label (5 classes per task) | Original Split | |
| | | Time-IL | 24 | Time (Publication year) | Random (4:1:5) | |
| | ogbn-mag | Task-IL, Class-IL | 128 | Class label (2 classes per task) | Original Split | |
| | | Time-IL | 10 | Time (Publication year) | Random (4:1:5) | |
| | CoraFull | Task-IL | 35 | Class label (2 classes per task) | Original Split | |
| | ogbn-products | Class-IL | 9 | Class label (5 classes per task) | Original Split | |
| | ogbn-proteins | Domain-IL | 8 | Domain (Species) | Random (4:1:5) | ROC-AUC |
| | Twitch | Domain-IL | 21 | Domain (Language) | Random (4:1:5) | Accuracy |
| Link Classification (LC) | Bitcoin-OTC | Task-IL, Class-IL | 3 | Class label (2 classes per task) | Random (8:1:1) | Accuracy |
| | | Time-IL | 7 | Time | | ROC-AUC |
| Link Prediction (LP) | ogbl-collab | Time-IL | 50 | Time (Publication year) | | Hits@50 |
| | Wiki-CS | Domain-IL | 54 | Domain (labels of endpoints) | | |
| | Ask-Ubuntu | Time-IL | 69 | Time (Interaction timestamp) | | |
| | Facebook | Domain-IL | 8 | Domain (Categories of pages) | | |
| Graph Classification (GC) | MNIST | Task-IL, Class-IL | 5 | Class label (2 classes per task) | Original Split | Accuracy |
| | CIFAR10 | Task-IL, Class-IL | 5 | Class label (2 classes per task) | Original Split | |
| | Aromaticity | Task-IL, Class-IL | 10 | Class label (3 classes per task) | Original Split | |
| | NYC-Taxi | Time-IL | 16 | Time (Month) | Random (6:2:2) | |
| | Sentiment140 | Time-IL | 11 | Time (Day) | | |
| | ogbg-ppa | Domain-IL | 11 | Domain (Species) | Random (8:1:1) | |
| | ogbg-molhiv | Domain-IL | 20 | Domain (Scaffold) | Random (8:1:1) | ROC-AUC |

of the edge. As labels, we used binary ones indicating whether each graph indicates taxi traffic on weekdays (Mon.-Fri.) or weekends (Sat.-Sun.). To construct the tasks, the generated graphs from 2018 to 2021 were divided chronologically into intervals of three months, resulting in 16 tasks.

For Sentiment140, we used the SpaCy (Honnibal et al., 2020) library to generate the graphs from online posts on Twitter, a social media platform. Specifically, we used the `en_core_web_lg` pipeline, which contains 514k unique word vectors of 300 dimensions. To sample posts, we first removed the posts containing at least one out-of-vocabulary token. Then, for each day from May 26 to July 7, 2009, we randomly sampled 250 posts with positive sentiment and 250 posts with negative sentiment. For every sampled post, using the pipeline, we obtained a graph (spec., tree) representing the dependency between the tokens (i.e., nodes) in the post and token embeddings (i.e., node features) To construct the tasks, we divide the graphs based on the dates of the corresponding posts.

For the remaining time-IL scenarios, we divided each dataset chronologically, leveraging time information in the raw data. For ogbn-arxiv, we constructed the first task with the paper published before the year 1998. For each subsequent $i$-th task, we used the papers published in the year $(1996 + i)$. Similarly, for ogbl-collab, we constructed the first task using the paper published before the year 1971. For each subsequent $i$-th task, we used the papers published in the year $(1969 + i)$. For ogbn-mag, we constructed each $i$-th task with the papers published in the year $(2009 + i)$. For Ask-Ubuntu, we used the interactions occurring within the same month to form each task.

**Statistics of the tasks.** Some task statistics for each benchmark scenario are available at https://anonymous.4open.science/r/BeGin-C4B0. For Task-IL and Class-IL, the class grouping procedure and thus task statistics depend on random seeds. Thus, we report the statistics when the random seed is fixed to 0.

### I.2 LICENSE OF ASSETS

**Implementation.** All assets of BEGIN are available under the Apache license 2.0. All assets we utilized from DGL (Wang, 2019) and DGL-LifeSci (Li et al., 2021) are also available under the Apache license 2.0. For the implementation of GEM, TWP, and ERGNN, we utilized the implementation from CGLB (Zhang et al., 2022b), which is available under CC BY-NC 4.0. For the implementation of ContinualGNN, we utilized the implementation written by the author, whose license is not specified.

**Benchmark Datasets.** The detailed license information for the considered datasets are as follows:

- The Cora, Citeseer, CoraFull, Bitcoin-OTC, and Wiki-CS datasets are publicly available. Wiki-CS is under the MIT license, but the licenses for these datasets are not specified. For accessing the datasets, we used DGL (Wang, 2019).

- The ogbn-arxiv, ogbn-mag, and ogbl-collab datasets are publicly available under the ODC-BY license. We used OGB (Hu et al., 2020) to access the datasets.

- The ogbn-products dataset is publicly available under the Amazon license. We used OGB (Hu et al., 2020) to access the datasets.

- The ogbn-proteins, ogbg-molhiv, and ogbg-ppa datasets are publicly available under the CC-0 license and the MIT license, respectively. We used OGB (Hu et al., 2020) to access the datasets.

- The Twitch and Facebook datasets are publicly available under the MIT license. The Ask-Ubuntu dataset is also publicly available, but the license is not specified. These three datasets are available at http://snap.stanford.edu/data/index.html.

- The MNIST and CIFAR10 datasets are publicly available under the MIT license. We used PyG (Fey & Lenssen, 2019) to access the datasets.

- The Aromaticity dataset, which is a subset of the PubChem BioAssay dataset (Xiong et al., 2019), is publicly available under the CC BY-NC 3.0 license. We used DGL-LifeSci (Li et al., 2021) to access the dataset.

- The raw data of NYC-Taxi is publicly available, and the license is not specified. We transformed the Yellow Taxi Trip data collected from 2018 to 2021 into graphs, as described in Section 3.3. The graphs are available in our framework, and some basic statistics are provided in Table 2.

- The raw data of Sentiment140 is publicly available, and the license is not specified. We transformed the online posts with sentiment into graphs, as described in Section 3.3. The graphs are available in our framework, and some basic statistics are provided in Table 2.

## J DETAILED COMPARISON WITH EXISTING STUDIES ON GRAPH CL

In this section, we describe how our benchmark scenarios and dataset usage differentiate from existing studies, focusing on three key aspects.

**Distinction in Definitions of Incremental Settings** Previous research on graph CL has often employed disparate settings under the same name, coupling various dimensions of change without sufficient justification. We systematically define four dimensions of changes (task, class, domain, and time) in graph CL and establish an incremental setting for each, without unnecessary coupling between them. For example, in several existing studies, changes in the input graph and tasks were coupled without sufficient justification or a clear relation to real-world graph evolution (Zhang et al., 2022b; Zhou & Cao, 2021).

Even among the seven studies conducted in the setting where the input graph changes as the task changes, there is a variation in how the input graph changes. For each task, three studies (Cai et al., 2022; Wang et al., 2022; Tan et al., 2022) utilized subgraphs only containing nodes available in the current task, while the other three studies (Sun et al., 2023; Zhang et al., 2022a; Zhou & Cao, 2021) used an evolving graph containing nodes observed in previous tasks. In the case of (Zhang et al., 2022b), experiments were conducted in both of these settings.

Table 11: **Coverage of scenarios.** 'N', 'L', and 'G' denote node classification, link prediction (classification), and graph classification, respectively. 'T', 'C', 'D', and 'Ti' denote task-, class-, domain-, and time-incremental settings, respectively. We compared ours with 14 studies on (knowledge) graph CL methods, including 2 existing benchmarks (Carta et al., 2021; Zhang et al., 2022b). We categorize the incremental settings based on our criteria. An asterisk (*) indicates that changes in the input graph and tasks were coupled without sufficient justification or a clear relation to real-world graph evolution, and a double asterisk (**) indicates that the design of the incremental setting is slightly different from ours.

| | (Graph Problem, Incremental Setting) | | | | | | | | | | | |
|---|---|---|---|---|---|---|---|---|---|---|---|---|
| | (N,T) | (N,C) | (N,D) | (N,Ti) | (L,T) | (L,C) | (L,D) | (L,Ti) | (G,T) | (G,C) | (G,D) | (G,Ti) |
| Liu et al. (2021) | O | | | | | | | | O | | | |
| Carta et al. (2021) | | | | | | | | | | O | | |
| Galke et al. (2021) | | | | O** | | | | | | | | |
| Cai et al. (2022) | O* | | | | | | | | | | | |
| Sun et al. (2023) | O* | | | | | | | | | | | |
| Zhang et al. (2022a) | | O* | | | | | | | | | | |
| Zhou & Cao (2021) | O* | | | | | | | | | | | |
| Wang et al. (2022) | | O | | | | | | | | | | |
| Daruna et al. (2021) | | | | | | O* | | | | | | |
| Zhang et al. (2022b) | O* | O* | | | | | | | O | O | | |
| Kim et al. (2022) | | | | O** | | | | | | | | |
| Tan et al. (2022) | | O* | | | | | | | | | | |
| Rakaraddi et al. (2022) | O | O | | | | | | | | | | |
| Han et al. (2020) | | | | | | | | | | | O | |
| **Ours** | **O** | **O** | **O** | **O** | **O** | **O** | **O** | **O** | **O** | **O** | **O** | **O** |

In addition, as shown in Table 11, our search yielded only one study that considers domain-IL settings within the domain of graph CL, and only two studies (Galke et al., 2021; Kim et al., 2022) consider the time-IL settings. Note that the settings of (Galke et al., 2021) and (Kim et al., 2022) differ from ours. In their settings, nodes from previous tasks were used for training, and the nodes in the current task were used for evaluation. However, these methods are less suitable for computing forgetting, compared to ours, since the labels of all nodes are eventually revealed during the training procedure.

**Distinction in Learning Problems** For each of our refined four incremental settings, we formulate commonly-considered graph learning problems (node-, link-, and graph-level), which result in 12 combinations. To ensure their utility in benchmarking graph continual learning, these problems are carefully and often non-trivially designed to highlight various aspects such as catastrophic forgetting, forward/backward transfer, and other key factors in continual learning methods. As shown in Table 11, our framework covers 5 more scenarios than the union of the coverage of existing studies. Except for Daruna et al. (2021), which is specialized to knowledge graphs, our search yielded no studies tackling link-level problems in graph CL. Moreover, our search yielded no study that considers domain-IL settings.

**Distinction in Datasets** As seen in Table 12, we newly used 11 datasets that have not been used in the previous Graph CL literature. Even for previously considered datasets, in most cases, we made better use of them under more incremental settings. For example, we used the ogbn-mag dataset for node classification under Time-IL settings and used the ogbg-ppa dataset for graph classification under Domain-IL settings. Furthermore, commonly utilized datasets, such as Cora, Citeseer, ogbn-arxiv, CoraFull, and Reddit, mainly focused on the limited combinations (spec. NC for Task-/Class-/Time-IL) of the problem levels and the incremental settings, and most of them are obtained from bibliographic data. BeGin provides 31 benchmark scenarios with the real-world graph datasets obtained from 9 application domains (citation, molecule, co-purchase, hyperlink, trading, co-authorship, image, traffic, and language) and covers all combinations of the problem levels and the incremental settings.

Table 12: **Comparison of datasets and incremental settings considered in each study.** 'T', 'C', 'D', and 'Ti' denote task-, class-, domain-, and time-incremental settings, respectively. An asterisk (*) indicates a dataset not used in this study. A double asterisk (**) indicates that the incremental tasks were constructed without actual time information, due to the absence of the information.

| Datasets | Liu et al. (2021) | Carta et al. (2021) | Galke et al. (2021) | Cai et al. (2022) | Sun et al. (2023) | Zhang et al. (2022a) | Zhou & Cao (2021) | Wang et al. (2022) | Zhang et al. (2022b) | Kim et al. (2022) | Tan et al. (2022) | Rakaraddi et al. (2022) | Han et al. (2020) | Ours |
|---|---|---|---|---|---|---|---|---|---|---|---|---|---|---|
| **Datasets Used for Node-level Problems** | | | | | | | | | | | | | | |
| Cora | | | | C | T | | T | C | | | | T,C | | T,C |
| Citeseer | | | | C | T | | T | C | | | | T,C | | T,C |
| Actor* (Pei et al., 2020) | | | | C | T | | T | C | | | | | | |
| CoraFull | T | | | | | | | | | T,C | | T,C | | T |
| Pubmed* (Sen et al., 2008) | | | | | | | | | | | Ti** | | | |
| DBLP* (Tang et al., 2008) | | | Ti | | | | | | C | | | | | |
| ogbn-arxiv | | | | C | T | | | | Ti | T,C | | | | T,C,Ti |
| Reddit (Hamilton et al.) | T | | | | T | | | | Ti | T,C | Ti | | | |
| PharmaBio* (Melnychuk et al., 2019) | | | Ti | | | | | | | | Ti | | | |
| ogbn-proteins | | | | | | | | | | | | | | D |
| Articles* (Cai et al., 2022) | | | | | | T | | | | | | | | |
| ogbn-mag | | | | | | | | | | | | | | T,C,Ti |
| PPI* (Zitnik & Leskovec, 2017) | T | | | | | | | | | | | | | |
| ogbn-products | | | | C | | | | C | Ti** | T,C | Ti** | | | C |
| Twitch | | | | | | T | | | | | | | | D |
| Amazon* (He & McAuley, 2016) | | | | | | | | | | | | | | |
| Amazon-computer* (McAuley et al., 2015b) | T | | | | | | | | | | | T,C | | |
| Amazon-clothing* (McAuley et al., 2015a) | | | | | | | | | C | | | | | |
| **Datasets Used for Link-level Problems** | | | | | | | | | | | | | | |
| Wiki-CS | | | | | | | | | | | | | | D |
| Bitcoin-OTC | | | | | | | | | | | | | | T,C,Ti |
| ogbl-collab | | | | | | | | | | | | | | Ti |
| Facebook | | | | | | | | | | | | | | D |
| Ask-Ubuntu | | | | | | | | | | | | | | Ti |
| **Datasets Used for Graph-level Problems** | | | | | | | | | | | | | | |
| MNIST | | C | | | | | | | | | | | | T,C |
| CIFAR10 | | C | | | | | | | | | | | | T,C |
| Aromaticity | | | | | | | | | | T,C | | | | T,C |
| ogbg-molhiv | | | | | | | | | | | | | | D |
| ogbg-ppa | | | | | | | | | | T | | | | D |
| SIDER* (Wu et al., 2018) | | C | | | | | | | | | | | | |
| NYC-Taxi | | | | | | | | | | | | | | Ti |
| Sentiment140 | | | | | | | | | | | | | | Ti |
| Tox21* (Huang et al., 2014) | T | | | | | | | | | T | | | | |
| PolitiFact* (Shu et al., 2020) | | | | | | | | | | | | | D | |
| GossipCop* (Shu et al., 2020) | | | | | | | | | | | | | D | |
| **Total # of Settings** | 5 | 3 | 2 | 5 | 5 | 2 | 3 | 4 | 5 | 12 | 4 | 8 | 2 | 31 |

