# OpenReview forum: "BeGin: Extensive Benchmark Scenarios and An Easy-to-use Framework for Graph Continual Learning"
_ICLR.cc/2024/Conference — ICLR 2024 Conference Withdrawn Submission_

### Official Review · Reviewer_mU8z · 2023-10-30

**Soundness:** 3 good
**Presentation:** 2 fair
**Contribution:** 3 good
**Rating:** 5
**Confidence:** 3

**Summary:**

This paper proposes a graph continual learning dataset and defines standard incremental settings and evaluations. It proposes a trainer to evaluate the performance.

**Strengths:**

1. The proposed datasets seem to be useful in graph incremental learning.
2. The dataset settings and evaluations seem to be sound.

**Weaknesses:**

1. The authors do not propose their method for graph incremental learning.
2. The organization of this paper is not good. The main contribution is the dataset but there is no intuitive information of what the dataset looks like. For example, what it contains, the scale, and other information.
3. The performance in Table 3 is not complete.

**Questions:**

see the weakness

---

> ### Author Response · Authors · 2023-11-18
> **Response to Reviewer mU8z (1)**
>
> We deeply appreciate your very constructive reviews.
>
> [**Weakness 1]** The authors do not propose their method for graph incremental learning.
>
> - Our work falls under the category of datasets and benchmarks, which is officially considered at ICLR. We believe that our contributions extend beyond just proposing a new method for graph CL.
>     - Indeed, there have been many benchmark works published in ICLR [1, 2, 3, 4]
> - In the below answer, we clarify our contributions distinguished from existing works.
> - **Clarification of our contributions**
>     - Firstly, to the best of our knowledge, we are the **FIRST** to introduce **parameter-isolation-based graph CL methods**, which perform overall the best under Task-IL settings.
>     - Additionally, we summarize our contributions through **THREE** aspects.
>         - **Definitions of incremental settings**
>             - Previous research on graph CL has often employed disparate settings under the same name, and has coupled various dimensions of change without sufficient justification. We **FIRST** systematically define four dimensions of changes (task, class, domain, and time) in graph CL and establish an incremental setting for each, **without unnecessary coupling between them**.
>                 - For example, in several existing studies, changes in the input graph and tasks were coupled without sufficient justification or a clear relation to real-world graph evolution.
>                 - Even among the seven studies conducted in the setting where the input graph changes as the task changes, **there is a variation in how the input graph changes**. For each task, three studies [9, 14, 16] utilized subgraphs only containing nodes available in the current task, while the other three studies [9, 10, 11] used an evolving graph containing nodes observed in previous tasks. In the case of [10], experiments were conducted in both of these settings.
>             - As summarized in `Table 11 of Appendix`, our search yielded **NO study that considers domain-IL settings** within the domain of graph CL, and only two studies [7, 15] consider the time-IL settings. Note that the settings of [7] and [15] differ from ours. In [7] and [15], nodes from previous tasks were used for training, and the nodes in the current task were used for evaluation. However, these methods are less suitable for computing forgetting, compared to ours, since the labels of all nodes are eventually revealed during the training procedure.
>         - **Broader learning problems**
>             - For each of our refined four incremental settings, we formulate commonly considered graph learning problems (node-, link-, and graph-level), which result in 12 combinations. To ensure their utility in benchmarking graph continual learning, these problems are carefully and often non-trivially designed to highlight various aspects such as catastrophic forgetting, forward/backward transfer, and other key factors in continual learning methods.
>             - As summarized in `Table 11 of Appendix`, except for [13], which is specialized to knowledge graphs, our search yielded **NO studies tackling link-level problems in graph CL**. Moreover, our search yielded **NO study that considers domain-IL settings**.
>             - Our framework **covers 5 more scenarios than the union of the coverage of existing studies** **[5-17],** as summarized in `Table 11 of Appendix`.
>         - **Extensive datasets & benchmark results**
>             - As summarized in `Table 12 of Appendix`, we newly used **11 datasets that have NOT been used in the previous graph CL literature [5-17]**.
>                 - `ogbn-proteins, ogbn-mag, Twitch, Wiki-CS, Bitcoin-OTC, ogbl-collab, Facebook, Ask-Ubuntu, ogbg-molhiv, NYC-Taxi,` and `Sentiment140`
>             - As summarized in `Table 12 of Appendix`, even for previously considered datasets, in most cases we made better use of them under **more incremental settings**.
>                 - Please note that we **did not claim to have created the datasets;** rather, we have provided clear references to their sources.

---

> ### Author Response · Authors · 2023-11-18
> **Response to Reviewer mU8z (2)**
>
> **[Weakness 2]** The organization of this paper is not good. The main contribution is the dataset but there is no intuitive information of what the dataset looks like. For example, what it contains, the scale, and other information.
>
> - As we clarified in our answer to Weakness 1, **our contributions are not focused only on the datasets**.
> - We have described the datasets in `Section 3.3 of the main paper.`
>     - For readers to easily understand the datasets and benchmark scenarios, we believe that it is essential to provide detailed and accurate descriptions of the incremental settings. Thus, we have covered the incremental settings in relatively great detail in our main paper, and due to page limitations, we were forced to address the dataset more briefly.
> - As a supplement to this, we provided much more comprehensive descriptions of the datasets and scenarios in `Table 10 of Appendix` and our documents [18,19].
>
> **[Weakness 3]** The performance in Table 3 is not complete.
>
> - Please refer to the answer to Weakness 1 of reviewer xQU7.
>
> ### References
>
> [1] Wu, Yuhuai, et al. "Int: An inequality benchmark for evaluating generalization in theorem proving." *arXiv preprint arXiv:2007.02924* (2020).
>
> [2] Schneider, Frank, Lukas Balles, and Philipp Hennig. "DeepOBS: A deep learning optimizer benchmark suite." *arXiv preprint arXiv:1903.05499* (2019).
>
> [3] Li, Wenda, et al. "Isarstep: a benchmark for high-level mathematical reasoning." *arXiv preprint arXiv:2006.09265* (2020).
>
> [4] Li, Chaojian, et al. "Hw-nas-bench: Hardware-aware neural architecture search benchmark." *arXiv preprint arXiv:2103.10584* (2021).
>
> [5] Liu, Huihui, Yiding Yang, and Xinchao Wang. "Overcoming catastrophic forgetting in graph neural networks." *Proceedings of the AAAI conference on artificial intelligence*. Vol. 35. No. 10. 2021.
>
> [6] Carta, Antonio, et al. "Catastrophic forgetting in deep graph networks: an introductory benchmark for graph classification." *arXiv preprint arXiv:2103.11750* (2021).
>
> [7] Galke, Lukas, et al. "Lifelong learning of graph neural networks for open-world node classification." *2021 International Joint Conference on Neural Networks (IJCNN)*. IEEE, 2021.
>
> [8] Cai, Jie, et al. "Multimodal continual graph learning with neural architecture search." *Proceedings of the ACM Web Conference 2022*. 2022.
>
> [9] Sun, Li, et al. "Self-supervised continual graph learning in adaptive riemannian spaces." *Proceedings of the AAAI Conference on Artificial Intelligence*. Vol. 37. No. 4. 2023.
>
> [10] Zhang, Xikun, Dongjin Song, and Dacheng Tao. "Hierarchical prototype networks for continual graph representation learning." *IEEE Transactions on Pattern Analysis and Machine Intelligence* 45.4 (2022): 4622-4636.
>
> [11] Zhou, Fan, and Chengtai Cao. "Overcoming catastrophic forgetting in graph neural networks with experience replay." *Proceedings of the AAAI Conference on Artificial Intelligence*. Vol. 35. No. 5. 2021.
>
> [12] Wang, Chen, et al. "Lifelong graph learning." *Proceedings of the IEEE/CVF conference on computer vision and pattern recognition*. 2022.
>
> [13] Daruna, Angel, et al. "Continual learning of knowledge graph embeddings." *IEEE Robotics and Automation Letters* 6.2 (2021): 1128-1135.
>
> [14] Zhang, Xikun, Dongjin Song, and Dacheng Tao. "Cglb: Benchmark tasks for continual graph learning." *Advances in Neural Information Processing Systems* 35 (2022): 13006-13021.
>
> [15] Seoyoon Kim, Seongjun Yun, and Jaewoo Kang. “Dygrain: An incremental learning framework for dynamic graphs”. International Joint Conference on Artificial Intelligence, 2022.
>
> [16] Tan, Zhen, et al. "Graph few-shot class-incremental learning." *Proceedings of the Fifteenth ACM International Conference on Web Search and Data Mining*. 2022.
>
> [17] Rakaraddi, Appan, et al. "Reinforced continual learning for graphs." *Proceedings of the 31st ACM International Conference on Information & Knowledge Management*. 2022.
>
> [18] https://anonymous.4open.science/r/BeGin-C4B0/task_statistics.pdf
>
> [19] https://anonymous-submission-24.github.io/index.html

---

### Official Review · Reviewer_xQU7 · 2023-10-31

**Soundness:** 2 fair
**Presentation:** 2 fair
**Contribution:** 2 fair
**Rating:** 3
**Confidence:** 4

**Summary:**

This paper introduces a benchmark specifically designed for continual learning (CL) on graph data, diverging from the conventional focus on independent data types like images and text. The framework utilizes 20 public datasets, segmented into multiple tasks tailored to various continual learning settings, including task-, class-, domain-, and time-incremental settings at node, edge, and graph levels. The performance of diverse CL methods under these settings is assessed using four evaluation metrics: average performance, average forgetting, intransigence, and forward transfer.

**Strengths:**

1.	The general motivation of building benchmarks for continual learning on graphs is interesting and practical for the graph learning community.
2.	There are some merits in the design of the experiments. Different settings are considered and evaluated.
3.	The code is well-organized and the documentation is relatively easy to understand.

**Weaknesses:**

1.	One major concern is the large incompleteness of the evaluation results(about one third of entries in tables are n/a). The author should be clear on why methods are not applicable. Besides, three methods (PackNet, Piggyback, HAT) only support the task-IL setting. Why does the author include them as part of baselines, since they are not generalizable?
2.	There are a few claims that are doubtful in terms of the comparison with exciting benchmarks. In Appendix G.3 and Table 9, the authors claim that the same implemented method has about on average 28.6% fewer lines of code. However, simply comparing the line counts is not meaningful to demonstrate its own strength. The reviewer suggests the author calculate the running time of each method implemented by the author and existing works for a fair comparison.
3.	The reviewer appreciated that the author stated the weakness in terms of the small number of tasks due to the limited labeled data. However, this is also an important contribution to the benchmarks (and differentiates from the existing works). Even a synthetic dataset is supposed to suffice.
4.	A few results seem problematic. For the methods ‘PackNet’ and ‘Piggyback’, the average forgetting on a few datasets is 0.000±0.000 for each number of tasks. The number of digits seems not enough to differentiate the results. Other reasons need to be discussed if available.

**Questions:**

Please see weakness 1 to 4.

**Details Of Ethics Concerns:**

N.A.

---

> ### Author Response · Authors · 2023-11-18
> **Response to Reviewer xQU7 (1)**
>
> Thank you for your thoughtful comments.
>
> **[Weakness 1]** One major concern is the large incompleteness of the evaluation results(about one third of entries in tables are n/a). The author should be clear on why methods are not applicable.
>
> - First, we would like to emphasize that many graph CL methods, as well as general CL methods for non-graph data, are designed for particular incremental setting(s) without necessarily considering their generalizability to other incremental settings.
> - Thus, we believe that it is common and natural for certain (graph) CL methods to not be universally applicable to all incremental settings.
> - **Clarification for N/A entries**
>     - We can divide the N/A results into two parts in `Tables 3 and 4 of our main paper`.
>         - Results of `ERGNN` and `CGNN` on `ogbg-proteins` and link-/graph-level problems
>             - `ERGNN` and `CGNN`, which use a replay buffer, save some nodes of each class that appeared in the previous tasks.
>             - Since the processes of selecting nodes are specified to the multi-class node classification, the two models are not applicable to multi-label binary and link-/graph-level problems.
>         - Results of parameter isolation-based methods on incremental settings except Task-IL
>             - Parameter-isolation-based methods are only applicable to the Task-IL setting, as they require task ID to operate (please refer to Section F of Appendix for the details of parameter-isolation-based methods).
>             - However, except for the Task-IL setting, it is prohibited to access the task ID during evaluation.
>
> **[Weakness 2]** Besides, three methods (PackNet, Piggyback, HAT) only support the task-IL setting. Why does the author include them as part of baselines, since they are not generalizable?
>
> - In this work, we would like to emphasize that we aim to include as many methods as possible in our baselines, not only those (graph) CL methods that generalize to all incremental settings.
> - We believe **this will be greatly beneficial to users developing graph CL methods for Task-IL**.
>
> **[Weakness 3]** There are a few claims that are doubtful in terms of the comparison with exciting benchmarks. In Appendix G.3 and Table 9, the authors claim that the same implemented method has about on average 28.6% fewer lines of code. However, simply comparing the line counts is not meaningful to demonstrate its own strength. The reviewer suggests the author calculate the running time of each method implemented by the author and existing works for a fair comparison.
>
> - Our response is two-fold.
>     - **Number of lines**
>         - We acknowledge that comparing the number of lines for implementation doesn't fully represent a framework's strength.
>         - However, we believe that needing fewer lines to implement the same (functionally equivalent) method somewhat demonstrates the framework's efficiency, which indicates its strength.
>     - **Running time**
>         - We think that the running time is largely dependent on the implementation of a user, not on the framework.
>         - Therefore, we do not compare the running time to existing works.

---

> ### Author Response · Authors · 2023-11-18
> **Response to Reviewer xQU7 (2)**
>
> **[Weakness 4]** The reviewer appreciated that the author stated the weakness in terms of the small number of tasks due to the limited labeled data. However, this is also an important contribution to the benchmarks (and differentiates from the existing works). Even a synthetic dataset is supposed to suffice.
>
> - **About synthetic datasets**
>     - To ensure the value of the benchmark in graph CL, it is crucial that these results accurately represent performance in real-world applications. Failure to do so could mislead many research efforts.
>     - Additionally, we believe that creating a graph with labels, features, and a realistic structure, that can effectively reflect real-world performance, presents an extremely challenging open problem.
>     - Consequently, this falls beyond the scope of our current research. However, we appreciate your valuable suggestion for future work.
> - **Differentiation from existing graph CL works**
>     - We would like to highlight our contributions are already distinguished from existing works in many aspects. (please refer to **our answer to Weakness 1 of the reviewer mU8z**).
>     - Also, we have strived to mitigate the mentioned limitation (the small # of tasks) as much as possible by utilizing real-world datasets.
>         - **Comparison of the maximum number of tasks to existing graph CL benchmark [1]**
>             - We support up to **128 tasks (`ogbn-mag`), which is 93 more tasks** than the existing graph CL benchmark [1].
> - We would appreciate it if the reviewer acknowledged that we admitted our limitations and have made efforts to address them as much as possible.
>
> **[Weakness 5]** A few results seem problematic. For the methods ‘PackNet’ and ‘Piggyback’, the average forgetting on a few datasets is 0.000±0.000 for each number of tasks. The number of digits seems not enough to differentiate the results. Other reasons need to be discussed if available.
>
> - PackNet and Piggyback store the trained parameters for each task and use these stored parameters during inference. Therefore, since they use the same parameters for inference on each task, there is no forgetting in performance.
> - Consequently, Average Forgetting (AF), which quantifies the forgetting of previous tasks, becomes zero in two models.
> - It is important to note that the AF for HAT, which is also one of the parameter-isolation-based methods, can be non-zero. This is because HAT updates some parameters for efficiency.
>
> ### References
>
> [1] Zhang, Xikun, Dongjin Song, and Dacheng Tao. "Cglb: Benchmark tasks for continual graph learning." *Advances in Neural Information Processing Systems* 35 (2022): 13006-13021.
>
> [2] Carta, Antonio, et al. "Catastrophic forgetting in deep graph networks: an introductory benchmark for graph classification." *arXiv preprint arXiv:2103.11750* (2021).

---

### Official Review · Reviewer_DN96 · 2023-11-01

**Soundness:** 3 good
**Presentation:** 3 good
**Contribution:** 3 good
**Rating:** 5
**Confidence:** 3

**Summary:**

This paper presents an extensive benchmark for graph continual learning. Currently, literatures in graph continual learning domain are lack of standards and inconsistent with the settings and evaluation dataset and metrics. This work aims to solve this problem by proposing a comprehensive evaluation benchmark with 31 scenarios based on 20 real world graphs. The work groups graph continual learning settings into four, including task-incremental, class-incremental, domain-incremental and time-incremental, and 3 levels of problems Moreover, this work provides the evaluation results of 10 graph CL methods using the provided benchmark.

**Strengths:**

Graph continual learning is a relatively unexplored field compared to continual learning research in other modalities. A primary obstacle is the lack of a standard evaluation protocol, including benchmark datasets, settings, and metrics. This paper addresses this gap, working diligently toward establishing such standards.

In this work, the authors systematically propose a benchmark designed to cover all possible scenarios in graph continual learning. Furthermore, for each scenario, results from 10 graph continual learning methods are provided, facilitating future comparisons and research.

The technical details supplied in the paper are comprehensive, and the codebase is well-documented, adding value to the paper and aiding in reproducibility.

**Weaknesses:**

While the authors have diligently developed a benchmark comprising 12 combinations of graph CL settings, the comprehensiveness of this benchmark in capturing the breadth of potential scenarios within graph CL problems remains somewhat ambiguous to me. Taking a different vantage point, such as data availability in the past, introduces unique scenarios. For instance, a situation where past data is inaccessible could arise, drawing parallels to the data-free setting observed in image CL problems [ref-1].

[ref-1] Always Be Dreaming: A New Approach for Data-Free Class-Incremental Learning, James Smith et al., ICCV 2021


Given the distinct nature of graph CL, scenarios involving incremental learning with the addition of new nodes typically presume a direct connection to the existing graph. This raises an intriguing question: what happens when one or more of these new nodes are not interconnected? Addressing and discussing this scenario would significantly enhance the scope and utility of this benchmark paper.

**Questions:**

It is not clear to me the motivating example shown in Figure 3 (there are changes in (a) the number of nodes, (b) the number of edges, (c) the number of classes (or domains), and (d) the distribution over classes) is covered by the proposed settings. Could you provide a detailed explanation to clarify this matter?

---

> ### Author Response · Authors · 2023-11-18
> **Response to Reviewer DN96**
>
> We sincerely appreciate your comments!
>
> **[Weakness 1]** While the authors have diligently developed a benchmark comprising 12 combinations of graph CL settings, the comprehensiveness of this benchmark in capturing the breadth of potential scenarios within graph CL problems remains somewhat ambiguous to me. Taking a different vantage point, such as data availability in the past, introduces unique scenarios. For instance, a situation where past data is inaccessible could arise, drawing parallels to the data-free setting observed in image CL problems [1].
>
> - In our understanding, the reviewer is pointing out that the scenarios of node-/link-level problems, despite varying tasks, inherently share information via the underlying graph.
> - This implies that we might not be considering situations where access to all past information is completely inaccessible.
> - However, the settings that (some of) past data are inaccessible are considered in our problems.
>     - In graph-level problems, similar to the CL on independent data (e.g., image), all graphs for training seen in past tasks are inaccessible in a current task.
>     - In link-level problems, some links for training (i.e., additional links) in past tasks are inaccessible in a current task.
>     - In all problems under Time-IL settings, (some of) past data are inaccessible in a current task.
>     - Please refer to Section 3.2 of our main paper for detailed definitions of the problems.
>
> **[Weakness 2]** Given the distinct nature of graph CL, scenarios involving incremental learning with the addition of new nodes typically presume a direct connection to the existing graph. This raises an intriguing question: what happens when one or more of these new nodes are not interconnected? Addressing and discussing this scenario would significantly enhance the scope and utility of this benchmark paper.
>
> - We do not presume a connection between newly emerging instances (e.g., nodes) and the existing graph in any scenario. Isolated nodes commonly appear in real-world graphs, including our benchmark datasets.
> - Most, if not all, graph learning models naturally address them by leveraging the information from each individual node. Therefore, we believe our current scenarios already encompass such situations.
>
> **[Question 1]** It is not clear to me the motivating example shown in Figure 3 (there are changes in (a) the number of nodes, (b) the number of edges, (c) the number of classes (or domains), and (d) the distribution over classes) is covered by the proposed settings. Could you provide a detailed explanation to clarify this matter?
>
> - We argue that the scenarios under **Time-IL settings cover the example of Figure 3**, showing changes in the four aspects.
> - In `Figure 4 of Appendix`, we illustrate the temporal changes in 6 **aspects (including 4 aspects the reviewer mentioned)** on `ogbn-arxiv`, which are considered in Time-IL settings.
> - **Detailed explanation**
>     - Taking the example of a citation network where each node is a research paper, as time progresses, many papers will be published, and citations among these papers will naturally increase.
>     - Furthermore, if new fields of study (new classes/domains of nodes) emerge, there could be a distribution shift with the existing classes.
>
> ### References
>
> [1] Always Be Dreaming: A New Approach for Data-Free Class-Incremental Learning, James Smith et al., ICCV 2021

---

> > ### Comment · Reviewer_DN96 · 2023-11-22
> > **Thanks for the response.**
> >
> > Thanks for the response. I acknowledge the authors' coverage of some data-free scenarios. My earlier comment or suggestion was aimed at encouraging the authors to more effectively organize these scenarios into a distinct and clear section as a dataset paper. Having reviewed the comments and feedback from other reviewers, who have expressed similar concerns regarding the presentation, I find it necessary to adjust my rating downwards.

---

> > > ### Author Response · Authors · 2023-11-23
> > > **Re-response to Reviewer DN96**
> > >
> > > Thank you for your response. We will incorporate it into the final version of our manuscript.

---

### Official Review · Reviewer_amhN · 2023-11-01

**Soundness:** 2 fair
**Presentation:** 2 fair
**Contribution:** 1 poor
**Rating:** 3
**Confidence:** 4

**Summary:**

The work is a benchmark for graph continual learning, which does assume the graph to learn is fixed. In other words, it is a setting that extends the continual learning/ lifelong learning to graph data.

**Strengths:**

1. This paper propose a easy-to-use framework for implementation of graph CL methods.

2. Some preliminary experimental results using the framework are provided.

**Weaknesses:**

1. This paper does not have any technical contribution, and is mostly dividing the existing public graphs into different tasks for graph continual learning, therefore may not be suitable for ICLR.

2. Even from the perspective of proposing new datasets/benchmark/evaluation, this paper has limited contribution. The proposed continual learning setting and splitting is same as standard continual learning on non-graph data, and the work in this paper to divide them is trivial.

3. As for the contribution of a new framework for implementation and evaluation for graph CL methods. It seems like the contribution is an easy-to-use software for facilitating coding, which is off the scope of AI research.

**Questions:**

1. What is the essential difference between CL and graph CL? Is simply an extension of CL methods to graph data?

2. There are also other works on graph CL, is the experiment settings in this paper consistent with them?

3. Among the datasets used in this paper, which are already used by graph CL works, which are new?

4. For the same datasets used by both this work and the other graph CL works, are the experimental settings the same? Are the results the same?

5. The methods experimented in this paper also include some works with public code, e.g. GEM. Can the proposed framework obtain reasonably similar results as their official implementation?

6. Can this framework be used on computer vision tasks?

---

> ### Author Response · Authors · 2023-11-18
> **Response to Reviewer amhN (1)**
>
> Thank you for your thoughtful comments.
>
> **[Weakness 1]** This paper does not have any technical contribution, and is mostly dividing the existing public graphs into different tasks for graph continual learning, therefore may not be suitable for ICLR.
>
> - **More datasets and broader incremental settings**
>     - While we acknowledge that the majority of our datasets are derived from existing sources, we would like to emphasize that we **newly used 11 datasets** that have NOT been used in 13 previous graph CL literatures and **supported the broader incremental settings** in most datasets (including previously considered datasets. Please refer to `Table 12 in Appendix` for the details of our analysis.
>         - The newly used datasets are `ogbn-proteins, ogbn-mag, Twitch, Wiki-CS, Bitcoin-OTC, ogbl-collab, Facebook, Ask-Ubuntu, ogbg-molhiv, NYC-Taxi,` and `Sentiment140`.
> - **Challenges in processing datasets**
>     - Also, we would like to emphasize that our work indeed involves some challenges and novel ideas in processing datasets. We discuss the details in the answer to **Weakness 2**.

---

> ### Author Response · Authors · 2023-11-18
> **Response to Reviewer amhN (2)**
>
> **[Weakness 2]** Even from the perspective of proposing new datasets/benchmark/evaluation, this paper has limited contribution. The proposed continual learning setting and splitting is same as standard continual learning on non-graph data, and the work in this paper to divide them is trivial.
>
> - For experts such as yourself, what we did can be seen as a direct adaptation of the incremental settings that have been considered in other domains (e.g., computer vision). However, our work **indeed involves some challenges and novel ideas,** summarized as follows:
>     - **1. Challenges and novelty in designing benchmark scenarios**
>         - We formulate commonly used graph learning problems (node-, link-, and graph-level) under each incremental setting.
>             - Specifically, among the 12 combinations, five have not been considered in previous studies (Please refer to `Table 11 in Appendix`). Especially, the exploration of link-level problems and domain-IL settings remains largely unexplored within this context.
>             - To ensure their utility in benchmarking graph continual learning, these problems are carefully and often non-trivially designed to highlight various aspects such as catastrophic forgetting.
>             - Please refer to `Section 3.2` and `Appendix J` for details.
>     - **2. Challenges and novelty in selecting and processing datasets**
>         - We contend that the task of **identifying suitable datasets** demands examination and analysis of existing datasets due to their requirements.
>             - **Task-IL and Class-IL Settings:** To generate the incremental settings, graph(s) must have a sufficient number of classes, and each class must have a sufficient number of samples belonging to it to form tasks. For example, it is hard to formulate task-IL and class-IL using graph(s) with ‘binary’ labels.
>             - **Domain-IL and Time-IL Settings:**  Graph(s) must contain not only the labels for formulating a classification problem but also additional information indicating domain or time information.
>         - **Preprocessing each dataset**, even when it is previously modeled as a graph, demands significant analysis and consideration to ensure its suitability as a graph continual learning benchmark. Here are some examples:
>             - **Semantic information for domain-IL:** For domain information, we aim to utilize semantic information  (e.g. broadcaster language for `Twitch`, and species information for `ogbn-proteins` and `ogbg-ppa`), rather than topology information simply derived from the graph structure. This often requires the processing of RAW data.
>             - **`ogbn-mag`:** To ensure that the number of instances in tasks is not extremely low, among 349 classes indicating fields of studies, we use the 257 classes with at least 10 nodes. They are divided into 128 groups for Task-IL.
>             - **`Bitcoin-OTC`:** To balance the number of instances among the tasks, we form 6 groups of links based on their weights: [−10, −9), [−9, 0), [3, 4), [4, 5), [5, 6), and [6, 10], and formulate three binary classification tasks.
>             - **`NYC-Taxi`:** We explicitly incorporate seasonality in the `NYCTaxi` (a total of 4 years) dataset under the time-IL setting by using time information. Specifically, we allocate data for three months to each task, resulting in a total of 16 tasks.
>         - Please refer to `Section 3.3` of the main paper and `Appendix I.1` for more examples of our efforts.
>     - **3. Challenges and novelty in designing the framework and algorithms**
>         - **Dependency between Instances and Fool-proofness:** Compared to continual learning with “independent” data, unintentional information leakage is more likely to occur in node- and link-level problems due to the “dependency” between tasks. This is because not only information associated with the current task can be used (for example for graph convolution operations) but also information from previous and future tasks because they are connected by edges. While the problem formulation is designed to prevent information leakage, users should remain vigilant about this issue. To mitigate the risk of potential errors, our framework processes the input for each task, ensuring that it retains only the necessary and accessible information.
>             - For example, for time-IL settings, our framework restricts access to the edges that appear in subsequent tasks, as well as information from neighboring nodes, and it also prohibits accessing labels of nodes that appear in later tasks.
>         - **Three New Graph CL Algorithms**: For the extensiveness of benchmark results, to the best of our knowledge, we are the first to introduce **parameter-isolation-based graph CL methods**, which perform overall the best under task-IL settings.
>         - For additional details, please refer to Section 4 of the main paper and Appendix F.

---

> ### Author Response · Authors · 2023-11-18
> **Response to Reviewer amhN (3)**
>
> **[Weakness 3]** As for the contribution of a new framework for implementation and evaluation for graph CL methods. It seems like the contribution is an easy-to-use software for facilitating coding, which is off the scope of AI research.
>
> - First, we demonstrate the advantages of our proposed framework BeGin.
>     - By utilizing our BeGin framework, researchers can expect the following three major benefits:
>         - **Comprehensive evaluation across diverse aspects**
>             - By implementing their algorithms using BeGin, researchers can effortlessly evaluate their algorithms across 31 benchmark scenarios.
>             - These scenarios include diverse incremental settings and learning problems across 20 datasets, evaluated using four different metrics.
>             - Refer to `Section 3` of the main paper for details.
>         - **Easy implementation**
>             - Researchers can implement their algorithms with considerably less effort, often requiring just a few lines of code (as demonstrated in `Tutorial of Appendix G.2`).
>             - This approach also minimizes the potential for errors due to the fool-proofness of BeGin.
>             - Refer to `Section 4` of the main paper for details.
>         - **Easy comparison**
>             - Users can readily compare their algorithms with up to 10 baselines under the **same** setting, eliminating the need to individually implement these baselines.
>             - Refer to `Section 5` of the main paper for details.
>     - Considering these advantages, we are confident that our framework will greatly facilitate the implementation, evaluation, and comparison of new graph CL algorithms. This, in turn, will foster the adoption of graph data for assessing continual learning algorithms. This contribution holds value for the fields of both graph learning and continual learning.
> - Also, we would like to clarify **what we are pursuing through this work** beyond facilitating coding with the proposed benchmark framework BeGin.
>     - **Goal 1. We believe that benchmarks for graph CL should be improved in terms of quantity, problem diversity, application fields, and challenges.**
>         - Our belief is rooted in data-centric AI principles. That is, we believe that the progression of graph CL can be accelerated by the availability of extensive and standardized benchmarks.
>         - While utilizing bibliographic data (Cora, Citeseer, etc.) to classify the field of study represents a valuable benchmark, there should be further opportunities for exploration, as partially explored in previous studies.
>     - **Goal 2. We also believe that evaluations across extensive scenarios should require minimal effort and remain reproducible.**
>     - Regarding **Goal 1**, we provide 31 benchmark scenarios with 20 real-world graph datasets (11 of 20 datasets are newly used for graph CL) obtained from 9 domains (citation, molecule, co-purchase, hyperlink, trading, co-authorship, image, traffic, and language) and cover all the 12 combinations of the problem levels and the incremental settings (Please refer to `Section 3.3` of the main paper).
>     - Regarding **Goal 2**, we developed our framework, BeGin (please refer to the above answer and `Section 4` of the main paper for the advantages of BeGin).

---

> ### Author Response · Authors · 2023-11-18
> **Response to Reviewer amhN (4)**
>
> **[Question 1]** What is the essential difference between CL and graph CL? Is simply an extension of CL methods to graph data?
>
> - In contrast to CL with independent data (e.g., images and text), graph CL also considers inter-entity connections (e.g., edges) inherent in the underlying graph, while CL only relies on the data itself.
> - The methods for CL can be extended to the graph data.
>     - Indeed, some methods in our paper are general CL methods, not for graph-structured data. Note that the methods cannot address the information of connections.
>
> **[Question 2]** There are also other works on graph CL, is the experiment settings in this paper consistent with them?
>
> - Please see our answer to **Questions 4 and 5**.
>
> **[Question 3]** Among the datasets used in this paper, which graph CL already uses works, which are new?
>
> - Please see our answer to **Weakness 1**.
>     - The newly used datasets are `ogbn-proteins, ogbn-mag, Twitch, Wiki-CS, Bitcoin-OTC, ogbl-collab, Facebook, Ask-Ubuntu, ogbg-molhiv, NYC-Taxi,` and `Sentiment140`.
>
> **[Questions 4 & 5]** For the same datasets used by both this work and the other graph CL works, are the experimental settings the same? Are the results the same? The methods experimented in this paper also include some works with public code, e.g. GEM. Can the proposed framework obtain reasonably similar results as their official implementation?
>
> - We are confident that if the experimental settings (e.g., incremental setting, data split) are identical, the same results can be obtained.
> - However, previous works often have considered settings that differ from one another and from ours. Please refer to `Section J of Appendix` for details.
>
> **[Question 6]** Can this framework be used on computer vision tasks?
>
> - We think that BeGin can be used for any continual tasks, including computer vision tasks, that can be reduced to graph-level problems.
>     - To the best of our knowledge, a significant number of existing studies, including [1-4], tackle computer vision tasks by graph neural networks after converting images to graphs.
>
> ### References
>
> [1] Sui, Yongduo, et al. "Causal attention for interpretable and generalizable graph classification." *Proceedings of the 28th ACM SIGKDD Conference on Knowledge Discovery and Data Mining*. 2022.
>
> [2] Cai, Shaofei, et al. "Rethinking graph neural architecture search from message-passing." *Proceedings of the IEEE/CVF Conference on Computer Vision and Pattern Recognition*. 2021.
>
> [3] Knyazev, Boris, et al. "Image classification with hierarchical multigraph networks." *arXiv preprint arXiv:1907.09000* (2019).
>
> [4] Vasudevan, Varun, et al. "Image classification using graph neural network and multiscale wavelet superpixels." *Pattern Recognition Letters* 166 (2023): 89-96.

---

> ### Comment · Reviewer_amhN · 2023-11-23
> **responses to the rebuttal**
>
> Thanks for the detailed rebuttal from the authors. After reading the rebuttal, I feel that the major concerns still exist.
>
> For example, as for the technical contribution, the authors only explain from the perspective of providing new datasets. The described challenge 'it is hard to formulate task-IL and class-IL using graph(s) with ‘binary’ labels.' is not very convincing either. In the two previous benchmarks cited in this paper, both these settings are studied. In a word, I still feel the explanations in the rebuttal do not provide much improvement over the original version.
>
> This work may be a good one for introducing the graph continual learning topic, but is not very suitable to appear in ICLR.

---

> > ### Author Response · Authors · 2023-11-23
> > **Re-response to Reviewer amhN**
> >
> > Thank you for replying.
> >
> > Q1. For example, as for the technical contribution, the authors only explain from the perspective of providing new datasets
> >
> > A1. We have also detailed the challenges we addressed and the novel ideas in our work “**in our response to Weakness 2”**, as well as we discussed our perspective on proposing a new dataset.
> >
> > Q2. The described challenge 'it is hard to formulate task-IL and class-IL using graph(s) with ‘binary’ labels.' is not very convincing either.
> >
> > A2. The described challenge is held by the definition of Task-IL and Class-IL provided in Section 3.2.
> >
> > - As provided in Section 3.2, the set of classes should vary with tasks, and they are often disjoint in Task-IL.
> > - In Class-IL, the set of classes should grow over tasks.
> >
> > Therefore, it is impossible to formulate the scenarios with binary labels.
> >
> > Q3. In the two previous benchmarks cited in this paper, both these settings are studied
> >
> > A3. To the best of our knowledge, we are confident that [1] and [2] do not consider Task-IL and Class-IL with binary label datasets in the same experimental settings as our work.
> >
> > - In [1], (if we understand correctly) the reviewer means SIDER-tIL and Tox21-tIL datasets for GC.
> >     - Note that SIDER-tIL and Tox21-tIL are binary “**multi-label”** datasets.
> >     - Thus, [1] can consider the two datasets.
> >     - We would like to emphasize that 'task' as defined in Task-IL of [1] differs from our definition. (See A2 for the details.)
> > - Also, to the best of our knowledge, in [2], there **is no dataset with binary labels (including multi-label)**.
> >
> > [1] Zhang, Xikun, Dongjin Song, and Dacheng Tao. "Cglb: Benchmark tasks for continual graph learning." *Advances in Neural Information Processing Systems* 35 (2022): 13006-13021.
> >
> > [2] Carta, Antonio, et al. "Catastrophic forgetting in deep graph networks: an introductory benchmark for graph classification." *arXiv preprint arXiv:2103.11750* (2021).

---

### Author Response · Authors · 2023-11-20
**Gratitude for Your Reviews and Open for Further Discussions**

Dear Reviewers,

We express our gratitude for your valuable time in reviewing our work. We are available to address any additional questions you may have regarding our work or rebuttal. Kindly inform us if there are any such concerns.

Sincerely,

The Authors.